# Refining Lake Volume Estimation and Critical Depth Identification for Enhanced GLOF Event Anticipation

Nazir Ahmed Bazai [1,2], Paul A. Carling [3*], Peng Cui [2,4*], Hao Wang [2,4], Zhang Guotao [4], Liu Dingzhu [4,5],[6], Javed Hassan [7]

[1] Key Laboratory of Mountain Hazards and Earth Surface Process/Institute of Mountain Hazards and Environment, Chinese Academy of Sciences (CAS), Chengdu, China
[2] China-Pakistan Joint Research Center on Earth Sciences, Chinese Academy of Sciences and HEC, Islamabad, Pakistan.
[3] Geography and Environmental Science, University of Southampton, Southampton SO17 1BJ, UK
[4] Institute of Geographic Sciences and Natural Resources Research, Chinese Academy of Sciences, Beijing, China
[5] Earth Surface Process Modelling, German Research Centre for Geosciences (GFZ), Potsdam, Germany
[6] National Disaster Reduction Centre of China, Ministry of Emergency Management, Beijing, China
[7] DTU Space, Technical University of Denmark, 2800 Kongens Lyngby, Denmark

*Correspondence to*: Paul A. Carling: p.a.carling@soton.ac.uk  and Peng Cui: pengcui@imde.ac.cn

**Abstract.** Climate change leads to changes in glacier mass balance, including steady advancements and surges that reposition the glacier snouts. Glacier advancement can dam proglacial meltwater lakes. Within the Karakoram and surrounding regions, the positive feedback of climate change has resulted in more frequent ice-dammed glacial lake outburst floods  (GLOFs), often facilitated by englacial conduits. However, the complex and multi-factor processes of conduit development are difficult to measure. Determining the lake depths that might trigger GLOFs and the numerical model specifications for breaching are challenging. Empirical estimates of lake volumes, along with field-based monitoring of lake levels and depths and the assessment of GLOF hazards, enable warnings and damage mitigation. Using historical data, remote sensing techniques, high-resolution imagery, cross-correlation feature-tracking, and field-based data, we identified the processes of lake formation, drainage timing, and triggering depth. We developed empirical approaches to determine lake volume and trigger water pressure leading to a GLOF. An albeit weak correlation between glacier surge velocity and lake volume reveals that glacier surge may play a crucial role in lake formation and thus controls the size and volume of the lake. Lake volume estimation involves geometric considerations of the lake basin shape. A GLOF becomes likely when the lake's normalized depth (*n'*) exceeds 0.60, equivalent to a typical water pressure on the dam face of 510 kPa. These field and remotely sensed findings not only offer valuable insights for early warning procedures in the Karakoram but also suggest that similar approaches might be effectively applied to other mountain environments worldwide where GLOFs pose a hazard.

## 1. Introduction

Globally, glacier shrinkage is a strikingly visible sign of climate change, as is an apparent increase in the number of glacier hazards such as avalanches (Byers, Somos-Valenzuela, Shugar, McGrath, Chand and Avtar, 2023; Kääb and Girod, 2023; Li, Cui, Hu, Lu, Guo, Wang, Wang, Wang and Zhou, 2024; You and Xu, 2022) and glacial lake outburst floods (GLOFs) (Bazai,

Cui, Carling, Wang, Hassan, Liu, Zhang and Jin, 2021; Bhambri, Hewitt, Kawishwar, Kumar, Verma, Tiwari and Misra, 2019; Emmer, 2017; Zheng, Allen, Bao, Ballesteros-Cánovas, Huss, Zhang, Li, Yuan, Jiang and Yu, 2021). However, within High Mountain Asia (HMA), particularly the Karakoram, Kunlun Shan, and Eastern Pamirs, the glaciers gained mass since 1970 (Berthier and Brun, 2019; Gardelle, Berthier and Arnaud, 2012; Kääb, Treichler, Nuth and Berthier, 2015; Minora, Bocchiola, D'Agata, Maragno, Mayer, Lambrecht, Mosconi, Vuillermoz, Senese and Compostella, 2013; Yao, Thompson, Mosbrugger, Zhang, Ma, Luo, Xu, Yang, Joswiak and Wang, 2012). This positive response to climate change may now be over (Jackson, Azam, Baral, Benestad and Brun, 2023), with many glaciers more recently displaying stability (Ali, Khan, Hassan, Qureshi and Bano, 2021) or retreat (Singh, Varade, de Vries, Adhikari, Rawat, Awasthi and Rawat, 2023). Nonetheless, the mass gain influenced glacier dynamic behaviours, with the Karakoram glaciers thickening, increasing glacier surges (Bazai, Cui, Carling, Wang, Hassan, Liu, Zhang and Jin, 2021; Bazai, Cui, Liu, Carling, Wang, Zhang, Li and Hassan, 2022; Mu, Gao and Liang, 2024), and advancing glacier termini throughout the region (Bhambri, Bolch, Kawishwar, Dobhal, Srivastava and Pratap, 2013; Bolch, Pieczonka, Mukherjee and Shea, 2017). This behaviour contrasts with neighboring regions with more sustained negative glacier mass budgets, such as the Himalaya, Hindukush, and Tibet (Bazai, Cui, Carling, Wang, Hassan, Liu, Zhang and Jin, 2021; Bolch, Pieczonka and Benn, 2011; Frey, Machguth, Huss, Huggel, Bajracharya, Bolch, Kulkarni, Linsbauer, Salzmann and Stoffel, 2014). In the latter areas, glaciers continue to shrink, thin, and reduce volume, showing no significant glacier advance (Dehecq, Gourmelen, Gardner, Brun, Goldberg, Nienow, Berthier, Vincent, Wagnon and Trouvé, 2019; Farinotti, Immerzeel, de Kok, Quincey and Dehecq, 2020; Yao, Thompson, Mosbrugger, Zhang, Ma, Luo, Xu, Yang, Joswiak and Wang, 2012). As a result, the increase in moraine lake formation has increased the number of GLOFs in the glacier-retreating regions (Yong, Nie, Yongwei, Sheng, Qiao, Liu, Linshan, Liu, Shiyin and Liu, 2017). However, in regions where ice mass has increased, glacier advance has prompted the rapid formation of ice-dammed lakes accompanied by sudden releases of meltwater originating from these lakes (Carling, 2013; Hewitt, 1982; Hewitt, 1998; Hewitt and Liu, 2010; Singh, Varade, de Vries, Adhikari, Rawat, Awasthi and Rawat, 2023). Although the number and size of GLOFs may decrease with progressive deglaciation (Veh, Lützow, Tamm, Luna, Hugonnet, Vogel, Geertsema, Clague and Korup, 2023), ice-dammed lake floods currently represent the dominant hazard in cryospheric regions (Veh, Lützow, Kharlamova, Petrakov, Hugonnet and Korup, 2022), comprising 70% of GLOFs through recorded history (Carrivick and Tweed, 2016). In contrast, moraine-dammed lakes contribute only 9% (with the remaining 16%, 3%, and 2 % associated with unknown dam types, volcanic activity, and bedrock failure, respectively) (Carrivick and Tweed, 2016). Specific details of the GLOF hazard for HMA have been compiled by Shrestha, Steiner, Shrestha, Dhungel, Joshi, Inglis, Ashraf, Wali, Walizada and Zhang (2023).

Herein, the focus is upon ice-dammed lakes. The mechanisms and frequency of ice dam GLOFs remain poorly understood, hindering accurate prediction (Bazai, Cui, Carling, Wang, Hassan, Liu, Zhang and Jin, 2021; Cook, Kougkoulos, Edwards, Dortch and Hoffmann, 2016; Harrison, Kargel, Huggel, Reynolds, Shugar, Betts, Glasser, Haritashya, Klimeš and Reinhardt, 2018; Richardson and Reynolds, 2000). Recent studies have investigated changes in frequency due to climate change (Rick,

McGrath, McCoy and Armstrong, 2023; Veh, Lützow, Tamm, Luna, Hugonnet, Vogel, Geertsema, Clague and Korup, 2023), and there are regional assessments of flood volume and hazards (Rick, McGrath, McCoy and Armstrong, 2023). Despite these efforts, understanding the drainage and predicting flood events from ice-dammed lakes remain challenging. Nonetheless, anticipating the risks associated with these events is crucial due to their potential to cause devastating impacts on human lives and livelihoods, ecosystems, infrastructure (*e.g.*, roads, bridges, hydropower systems), river channel stability, and effects on agriculture and fisheries (Carrivick and Tweed, 2016; Cook, Kougkoulos, Edwards, Dortch and Hoffmann, 2016; Emmer, 2017; John, Clague, and, Stephen and Evans, 2000; Neupane, Chen and Cao, 2019; Zhang, Li, East, Walling, Lane, Overeem, Beylich, Koppes and Lu, 2022). GLOFs have been recorded up to 500 km from ice-dammed lakes (Hewitt and Liu, 2010), resulting in hundreds of human fatalities and the other impacts noted above (Carrivick and Tweed, 2016; Cui, Chen, Xiang and Su, 2014; Cui, Su, Zou, Chen and Zhang, 2015; Kreutzmann, 1994; Mason, 1929; Stuart-Smith, Roe, Li and Allen, 2021; Zhang, 1990; Zheng, Allen, Bao, Ballesteros-Cánovas, Huss, Zhang, Li, Yuan, Jiang and Yu, 2021).

Whilst progress has been made to understand the breaching mechanisms of moraine lake outburst floods, triggered by ice or debris falls, strong earthquake shaking, internal piping, or overtopping waves that exceed the shear resistance of the dam (Emmer and Vilímek, 2013; Richardson and Reynolds, 2000), the understanding of the mechanisms of ice dam lake outburst floods remains a challenge (Werder, Bauder, Funk and Keusen, 2010), making prediction using numerical modelling currently impossible. Therefore, there is an urgent need for simplified approaches to GLOF prediction to mitigate downstream impacts.

Despite the uncertainty related to the detail of GLOF initiation, sudden glacier advances during surge cycles have a prominent role in the formation of ice-dammed lakes by creating an ice barrier in the valleys, particularly at narrow valley floor sections and at confluences (Bazai, Cui, Carling, Wang, Hassan, Liu, Zhang and Jin, 2021; Bhambri, Hewitt, Kawishwar, Kumar, Verma, Tiwari and Misra, 2019), damming rivers (Singh, Varade, de Vries, Adhikari, Rawat, Awasthi and Rawat, 2023). Glacier surges have resulted in the formation of ice-dammed lakes in the Swiss Alps (Haeberli, 1983), Northern Norway (Xu, Bogen, Wang, Bønsnes and Gytri, 2015), Argentinian Patagonia (Vandekerkhove, 2021), Alaska (Trabant, March and Thomas, 2003), Karakoram, and in the Pamir (Bazai, Cui, Carling, Wang, Hassan, Liu, Zhang and Jin, 2021; Hewitt and Liu, 2010) and Tianshan regions (Ng, 2007; Shangguan, Ding, Liu, Xie, Pieczonka, Xu and Moldobekov, 2017). Recent studies reveal that the draining processes of ice-dammed lakes potentially involve one or more mechanisms: subglacial breaching, overspill, rapid ice mass instability, and slow deformation of subglacial cavities (Björnsson, 2003; Haemmig, Huss, Keusen, Hess, Wegmüller, Ao and Kulubayi, 2014; Round, Leinss, Huss, Haemmig and Hajnsek, 2017). Several attempts have been made to explore the drainage behaviour of ice-dammed lake outburst floods (Hewitt and Liu, 2010). However, due to the remoteness, danger, and inhospitable terrain where such lakes can be found, real-time data are few, and significant gaps remain in our knowledge of these processes.

In the Karakoram, ice-dammed lakes are found in five major valleys, three of which are densely populated and highly vulnerable to unexpected GLOFs. Recent advances in understanding have been made (Bazai, Cui, Carling, Wang, Hassan, Liu, Zhang and Jin, 2021) concerning the formation of episodic ice-dammed lakes, which, due to ice mass transfer variations, are linked to the changes in the glacier surface velocity, ice thickness (Singh, Varade, de Vries, Adhikari, Rawat, Awasthi and Rawat, 2023) and fluctuations in the crevasse density during the surge cycle (Rea and Evans, 2011; Sharp, 1985). Consequently, herein, we explore two main hypotheses: 1) that lake volume is related to glacier velocity, and 2) that there is a critical lake depth associated with ensuing GLOFs (Thoraninsson, 1939). As lake volume can dictate the characteristics of a GLOF, a third secondary hypothesis was addressed: 3) that ice-dammed lakes can exhibit geometries similar to regular geometric shapes, such that in the absence of detailed lake volume data, lake volumes might be estimated from geometric consideration. Despite advancements in knowledge, globally, the techniques for measuring and estimating the volume of the lake before an outburst, the critical depth (for GLOF release), and timely prediction of GLOFs remain largely unexplored or unidentified (Round, Leinss, Huss, Haemmig and Hajnsek, 2017; Shangguan, Liu, Ding, Guo, Xu, Xu and Jiang, 2016; Steiner, Kraaijenbrink, Jiduc and Immerzeel, 2018). Very limited ice-dammed lake volume data are available. These ice-dam lake volumes were measured either while the lake basin was empty (after a GLOF event) or partially filled and thus shallow (Round, Leinss, Huss, Haemmig and Hajnsek, 2017; Shangguan, Liu, Ding, Guo, Xu, Xu and Jiang, 2016; Steiner, Kraaijenbrink, Jiduc and Immerzeel, 2018). Given that their potential full volumes are unknown, the downstream threat from such lakes remains high. To measure the flood volume and flood magnitude for a deep and potentially full lake, the lake volume measurement is recognized as a critical variable that needs to be accurately calculated or at least well-estimated (Bazai, Cui, Carling, Wang, Hassan, Liu, Zhang and Jin, 2021; Bazai, Cui, Liu, Carling, Wang, Zhang, Li and Hassan, 2022). An accurate estimate of lake volume will also help explore the timing, triggering depth of the lake, and frequency of ice-dammed lake outburst floods in relation to surge cycles. Timing information can be approximated by correlating glacier velocities and GLOF occurrences (Bazai, Cui, Carling, Wang, Hassan, Liu, Zhang and Jin, 2021; Bazai, Cui, Liu, Carling, Wang, Zhang, Li and Hassan, 2022), which should assist in timely hazard assessment. Herein, the primary objective of this study is to enhance predictive capabilities regarding GLOF event timing by refining empirical lake volume estimation and identifying critical depths for future hazard and risk reduction. We seek to achieve the objective within a framework of adjustment of lake volume to glacier surge speed, which has implications for changes in the depth of lakes relative to the heights of the ice barriers that impound the lakes.

## 2. Study Area

The Karakoram Mountain ranges in HMA are known for their complex geology, climatic variability, and denudation processes, including debris flows, mudflows, landslides, rockfalls, avalanches, and GLOFs. As was noted in the preceding section, changes in glacier dynamics, increasing glacier surges, and a trend of increases in GLOF-related disasters characterize this region. These hazards are responsible for substantial economic losses, including the destruction of

residences, infrastructure such as roads and bridges, and agricultural areas, as well as blockages of transportation routes like the Karakoram Highway and other expressways (Shrestha, Steiner, Shrestha, Dhungel, Joshi, Inglis, Ashraf, Wali, Walizada and Zhang, 2023).

Glacier surges in the region have been recorded since the 15th century (Bazai, Cui, Carling, Wang, Hassan, Liu, Zhang and Jin, 2021). Since the application of remote sensing to the monitoring of the glaciers from 1970 to 2020, an increasing occurrence of glacier surges has been recorded from the 1990s, with some glacier surges being linked to the formation of ice-dammed lakes and subsequent GLOFs. Some lakes persist only seasonally, forming in the winter when temperatures are very low and draining slowly in the spring or summer. Other lakes are more persistent (Bhambri, Hewitt, Kawishwar, Kumar, Verma, Tiwari and Misra, 2019; Hewitt and Liu, 2010) and pose the potential for catastrophic outbursts. The most frequent glacier surges and formation of lakes leading to outburst floods in the Karakoram region occur for the Khurdopin, Kyager, and Shishper glaciers. For example, Round, Leinss, Huss, Haemmig and Hajnsek (2017) concluded that surges were the main factor controlling the formation of ice-dammed lakes associated with the Kyagar Glacier, with the volumes of the lakes reaching a maximum three years after the surge period (Li, Lv, Quincey, Taylor, Li, Yan, Sun and Guo, 2023). Similarly, Bazai, Cui, Liu, Carling, Wang, Zhang, Li and Hassan (2022) concluded that surge velocities have a significant effect of lake formation related to the Khurdopin Glacier.

Although the foreland of the Kyager Glacier, situated in the Shaksgam Valley, is uninhabited, GLOFs have caused damage and losses further downstream. Conversely, GLOFs from the Khurdopin and Shishper glaciers, located in the densely populated Hunza area, have resulted in casualties and substantial economic losses. Consequently, these glaciers and their lakes are selected for study. The focus of the broader investigation is to obtain the data necessary to understand the complex behavior of the glaciers and their drainage systems with a view to anticipating when the occurrence of GLOFs is imminent. Thus, there is an urgent need to identify trigger factors for GLOFs to provide downstream warnings in a timely fashion. A better understanding of the complex process behaviors should eventually lead to improved prediction of such events, not only within the Karakoram but also worldwide.

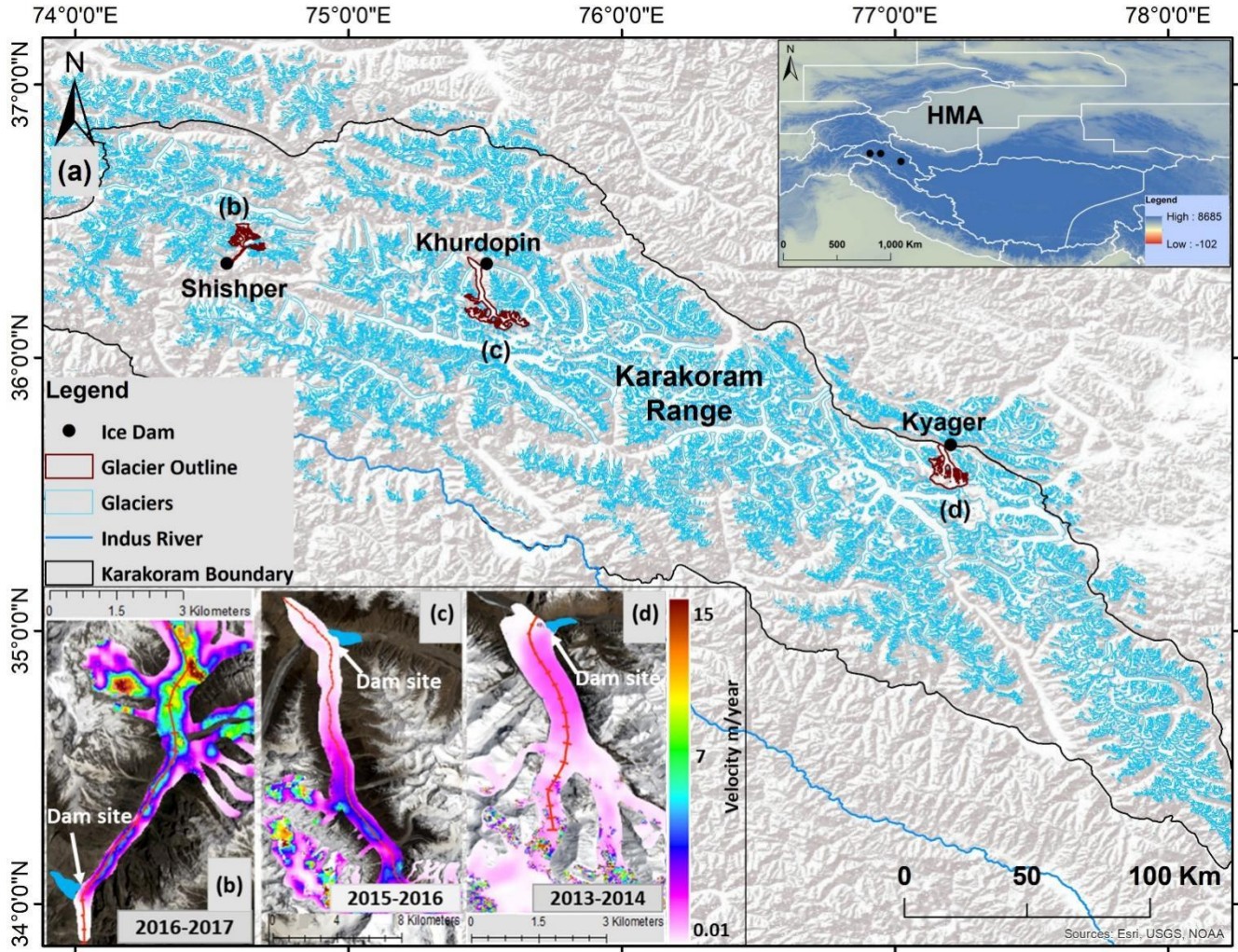

**Figure 1:** Overview of the study site in the Karakoram (a) and the High Mountain Asia (HMA) region; panels (b-d) present the extent of each glacier at a given time that surge speed has led to ice-dammed lake formation. The associated ice flow velocities are indicated. The background of panels (b-d) are Google Earth© images.

## 3. Data and Methods:

### 3.1. Remote sensing data

The identification and mapping of the Khurdopin, Kyager, and Shishper ice-dammed lakes were accomplished using open and commercial satellite imagery sources from 1970 to 2022. The datasets include 590 images of Landsat 2-5, 7-9, and 45 images from Sentinel-2, downloaded from the United States Geological Survey (USGS) website (http://earthexplorer.usgs.gov/) (Table S1). The commercial high-resolution images consisted of 35 images from Gaofen-1

(GF-1) and Gaofen-2 (GF-2), 11 images from SPOT-6 and SPOT-7 and five images from Global Planet (https://data.cresda.cn/#/2dMap, https://earth.esa.int/eogateway and https://www.planet.com/products/planet-imagery, respectively). The following DEM datasets have been used to estimate lake volume, depth and dam height: the Advanced Spaceborne Thermal Emission and Reflection Radiometer (ASTER) and the Phased Array type L-band Synthetic Aperture Radar (PALSAR)-DEM data scenes from the National Aeronautics and Space Administration (NASA) Earth Science Data

Center website (https://search.earthdata.nasa.gov/). KH-9 and Shuttle Radar Topography Mission (SRTM) data downloaded from http://earthexplorer.usgs.gov/ (Table S2). Field surveys of the Shishper glacier lakes were conducted in 2019, 2021, and 2022 and for Khurdopin in 2017 and 2018 using hand-held GPS and Uncrewed Aerial Vehicles (UAV) (see section 3.2) to determine annual lake extents, lake depths, glacier altitudes, and thickness, termini positions, and glacier surface displacements. The purpose of the field campaigns was to obtain: i) data on processes that could not be derived from remote

sensing and; ii) to obtain field data to calibrate/validate remote sensing-derived data. The glacier outlines were obtained from the Randolph Glacier Inventory (RGI 6.0) (Consortium, 2017) and modified according to surge movements with time (https://www.planet.com/products/planet-imagery/).

## 3.2. Glacier lake surface area mapping and glacier surface velocity

Satellite imagery had a spatial resolution of 0.8 to 30 m (Table S1). The use of high-resolution imagery aims to obtain

accurate lake surface levels. The images were selected based on the visibility of the glacier surface and lake areas, and overall, 23 ice-dammed lakes from eight surge events were identified related to the Khurdopin, Kyager, and Shishper glaciers (Table 1). The presence of lakes was determined based on the Normalized Difference Water Index (NDWI) (McFeeters, 1996), and the outlines of all 23 lakes were digitized manually using Landsat false-color composites (near-infrared, red, and green bands) to distinguish water bodies from other objects (Huggel, KääB, Haeberli, Teysseire and Paul,

2002). The extent of six Shishper and Khurdopin lakes that occurred after 2017 were obtained in the field using GPS (G639; Accuracy: Single: 1 ~ 3m; SBAS: 0.6m) survey points along the lake shorelines (Fig.2a-d), as well as from Uncrewed Aerial Vehicle (UAV) generated Digital Surface Models (DSMs). Alternatively, high-resolution satellite imagery from Planet (3 m) and GF-1 and 2 (0.8 m to 4 m resolution, respectively) and SPOT-6 and SPOT-7 (1.5 m) were used to extract the lake boundaries. The coupled lake extent and outlines help reduce the uncertainty of the lake extent obtained for Landsat 2-5

images. The above method was used to extract the extent of the previous Khurdopin and Kyager glacier lakes previously reported (Bazai, Cui, Carling, Wang, Hassan, Liu, Zhang and Jin, 2021; Bazai, Cui, Liu, Carling, Wang, Zhang, Li and Hassan, 2022), which data are incorporated into the current analysis.

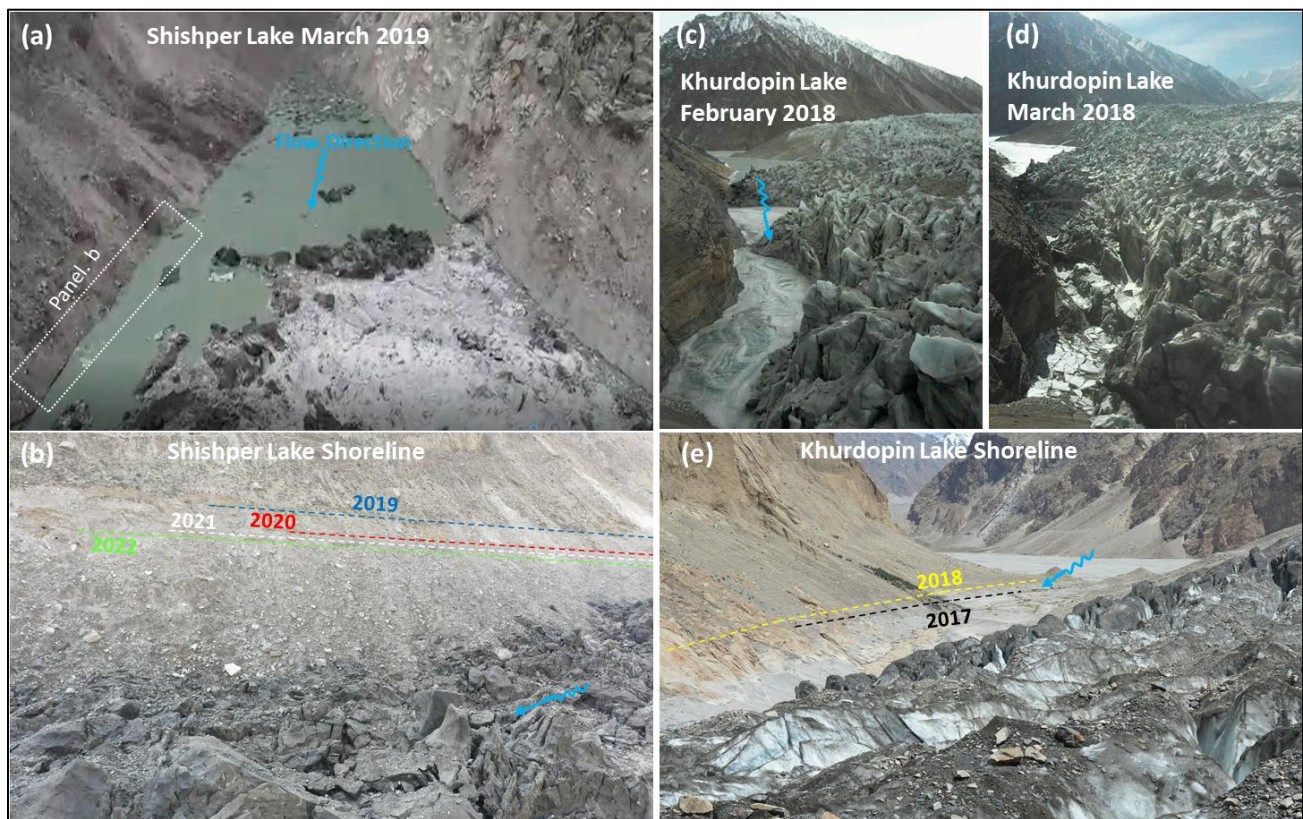

**Figure 2:** Shishper and Khurdopin glacial lake views in the field. a) Oblique view from a helicopter in March 2019 (image captured during lake monitoring by Gilgit-Baltistan Disaster Management Authority); b) the Shishper shoreline elevations of four lakes that outburst in the given years; c) and d) successive oblique views of the Khurdopin lake in the field; e) Khurdopin lake elevations in the given years. Wiggly blue lines are flow directions.

The Khurdopin, Kyager, and Shishper glaciers are surge-type glaciers (Copland, Sylvestre, Bishop, Shroder, Seong, Owen, Bush and Kamp, 2011; Hewitt, 1998). Since 1972, eight surge events have occurred from Khurdopin (three surges), Kyager (three surges), and Shishper (two surges) (Table 1) at an interval of 17–20 years for each glacier. The Landsat 2-4 images from 1970 to 1990 have errors in the selected glacier area. Therefore, the initial surges for Khurdopin and Kyager between 1970 and 1989 were not considered when estimating the annual velocity. Orthorectified Landsat scenes from TM to OLI–2 and Sentinel 2 were used to estimate the yearly and event-based velocities of all three glaciers from 1989 to 2022 to obtain information about the surge events and glacier front changes. Within this period, cloud-free images were chosen each year, although some satellite images were absent. Glacier velocities were recorded as annual averages, although daily measurements of glacier velocity were also determined to assess any effect on lake volume, given the possible velocity sensitivity to the triggering time of GLOFs.

The surface velocities were extracted along the central line of the Khurdopin, Shishper, and Kyager glaciers, highlighting the quiescent and surge phases obtained from published data (Bazai, Cui, Carling, Wang, Hassan, Liu, Zhang and Jin, 2021; Bazai, Cui, Liu, Carling, Wang, Zhang, Li and Hassan, 2022) using image-to-image correlation open-source software COSI-Corr (Leprince, Avouac and Ayoub, 2012; Leprince, Barbot, Ayoub and Avouac, 2007). The software effectively assesses the glacier surface velocity (Leprince, Avouac and Ayoub, 2012; Steiner, Kraaijenbrink, Jiduc and Immerzeel, 2018). Utilizing a displacement calculation, this technique was used to co-register and correlate surface features (Bazai, Cui, Carling, Wang, Hassan, Liu, Zhang and Jin, 2021; Steiner, Kraaijenbrink, Jiduc and Immerzeel, 2018). The surface velocity and overall movement during the surge were measured by observing changes in the GPS-registered glacier front positions every three months from March 2019 and measured for three years for the Shishper glacier in the field as well as for the Khurdopin glacier after six from June 2017 to July 2019. When coupled with COSI-Corr measured velocities, these latter procedures gave accurate results. The velocity estimation procedure generally yields an accuracy of ¼ of a pixel (Sattar, Goswami, Kulkarni and Das, 2019). Velocity root-mean-square errors (RMSE) were assessed to justify image processing accuracy. Examples of output are given in Fig. 1b to d.

### 3.3. Field observation and lake volume measurement

Six lakes were regularly monitored: four from the Shishper Glacier and two from the Khurdopin Glacier. Data for 23 GLOF events from eight surge cycles that occurred during the first year of each surge, or following, are presented in Table 1, with lakes resealing after each GLOF. The field data for six events from the Khurdopin and Shishper glaciers helped to reduce the uncertainty or validated data for 17 lakes for which data were obtained through remote sensing techniques (as explained in section 3.2).

For the Kyager Glacier, Li, Lv, Quincey, Taylor, Li, Yan, Sun and Guo (2023) suspected that the drainage conduit may not have been at the deepest part of the lake basin and its configuration changed between GLOF events. All the previously recorded lakes from Khurdopin and Shishper were drained via single subglacial conduits with stable inlet positions (i.e., the inlet was seen to be in the same place on each occasion) and varying outlet positions and conduit lengths. So, as closely as possible, we identified the inlet and outlet positions of the drainage conduits. As is shown within the Results, the inlet position of the conduit in the ice-dammed lake basin was always in the deepest position. The lowest ice dam height also tended to be in the vicinity of the conduit. The conduit inlet positions were geolocated within the empty lake basins using GPS, and the lake depths were calculated for these locations with reference to shoreline elevations (Table 1 and Figure 2). From the field survey, we noted that the presence of alignments of surface depressions in the glacier indicated the approximate position of curvilinear conduits, from which we estimated the conduit lengths between the inlet and outlet.

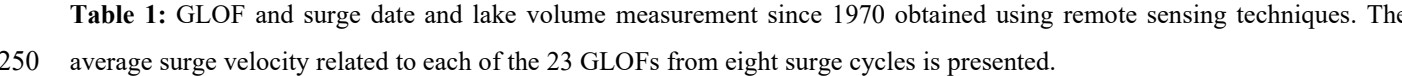

**Table 1:** GLOF and surge date and lake volume measurement since 1970 obtained using remote sensing techniques. The average surge velocity related to each of the 23 GLOFs from eight surge cycles is presented.

| No. | Glacier Name | Date | Sensor | Lake surface Elevation (m) | Lake Area km² | Aster/UAV Lake Volume Estimate (10⁶ m³) | Lake Vol Uncertainty (+/-10⁶ m³) | Vol. after [10⁶ m³] | Average velocity (m/d) | Date of next clear image after GLOF | Surge cycle and Resealed GLOF | Surge Duration in months |
|---|---|---|---|---|---|---|---|---|---|---|---|---|
| | Khurdopin | 20/08/1977 | LM02 | | | | | | | | 1977–1979 | May 1977– Aug 1979; 27 Months |
| 1 | Khurdopin | 15/8/1999 | | | | | | | 0.33 | | 1998-1999 | Jan 1995 to Sep 2002; 92 Months |
| 2 | Khurdopin | 05/30/2000 | L5 TM | 3440 | 1.87 | 186 | 2.1 | x | 0.33 | 08/26/2000 | Resealed | |
| | Khurdopin | 04/07/2001 | LE07 | 3416 | 0.295 | 19.5 | 1.5 | x | 0.44 | 06/26/2001 | Resealed | |
| 4 | Khurdopin | 07/15/2002 | LE07 | 3420 | 0.60 | 52.1 | 1.6 | 2.2 | 0.87 | 08/16/2002 | Resealed | |
| 5 | Khurdopin | 07/28/2017 | LE07 | 3415 | 0.180 | 16.2 | 1.4 | x | 1.41 | LC 08 08/01/2017 | 2016-2018 | June 2006 to Aug 2009: 38 months |
| 6 | Khurdopin | 03/18/2018 | LC08 | 3418 | 0.402 | 19.8 | 0.9 | x | 0.53 | 02/25/2018 | Resealed | |
| 7 | Kyager | 08/01/1977 | LM02 | 4785 | 1.181 | 40.73 | 5.8 | x | | 10/14/1977 | 1976-1977 | Jan 1975 to Aug 1978; 43 Months |
| 8 | Kyager | 07/18/1978 | LM02 | 4810 | 2.17 | 82.12 | 15.6 | x | | 06/07/1979 | Resealed | |
| 9 | Kyager | 03/08/1997 | L5 TM | 4823 | 3.30 | 127.3 | 2.9 | x | 0.4 | 04/09/1997 | 1994-1996 | Jan 1995 to Sep 2002; 92 Months |
| 10 | Kyager | 09/10/1998 | L5 TM | 4825 | 3.32 | 133.5 | 3.5 | x | 0.3 | 10/11/1998 | Resealed | |
| 11 | Kyager | 09/07/1999 | L7ETM+ | 4813 | 2.19 | 86.12 | 1.23 | x | 0.46 | 08/17/1999 | Resealed | |
| 12 | Kyager | 06/25/2000 | L7ETM+ | 4778 | 0.91 | 23.48 | 1.12 | x | 0.49 | 08/03/2000 | Resealed | |
| 13 | Kyager | 09/08/2002 | L7ETM+ | 4819 | 2.93 | 115.19 | 1.09 | x | 1.29 | 10/09/2002 | Resealed | |
| 14 | Kyager | 06/14/2008 | L5 TM | 4811 | 1.45 | 94.95 | 1.65 | x | 0.61 | 23/06/2008 | Resealed | June 2006 to Aug 2009: 38 months |
| 15 | Kyager | 07/28/2009 | L5 TM | 4808 | 1.39 | 91.35 | 1.56 | x | 0.56 | 04/08/2009 | Resealed | |
| 16 | Kyager | 07/16/2015 | L8 OLI | 4800 | 1.56 | 53.5 | 0.87 | x | 1.14 | 05/08/2015 | Resealed | Jan 2013 to Aug 2018: 67 months |
| 17 | Kyager | 07/14/2016 | L7ETM+ | 4804 | 1.63 | 45.89 | 1.49 | 2.9 | 0.53 | 30/07/2016 | 2014-2016 | |
| 18 | Kyager | 07/31/2016 | L7ETM+ | 4806 | 1.48 | 44.32 | 1.23 | x | 0.38 | 08/09/2016 | Resealed | |
| 19 | Kyager | 08/10/2017 | L8 OLI | 4815 | 2.91 | 113.99 | 0.73 | 11.9 | 0.38 | 26/08/2017 | Resealed | |
| 20 | Kyager | 08/06/2018 | L8 OLI | 4807 | 2.38 | 87.98 | 0.53 | x | 0.29 | 08/29/2018 | Resealed | |
| 21 | Shishper | 06/23/2019 | L8 OLI | 2650 | 0.37 | 24.10 | 2.1 | x | 0.95 | 07/13/2019 | C | 2017-2019 |
| 22 | Shishper | 05/29/2020 | L8 OLI | 2636 | 0.50 | 24.90 | 1.5 | x | 0.46 | 06/22/2020 | C | Resealed |
| 23 | Shishper | 05/16/2021 | L8 OLI | 2638 | 0.52 | 25.77 | 1.4 | x | 0.29 | 07/15/2020 | C | Resealed |

Dec 2018 to June 2022: 41 Months

| Shishper | 05/07/2022 | L8 OLI | 2641 | 0.41 | 27.66 | 1.1 | x | 0.24 | 21 05/10/20 22 | C | Resealed |

In addition, we used a UAV (DJI Mavic 2 Pro) equipped with a high-resolution camera (4000 pixels × 2250 pixels) to obtain multiple aerial photographs with a minimum of 85% image overlap (Entwistle and Heritage, 2017; Entwistle and Heritage, 2019; Tonkin and Midgley, 2016). The UAV flew at a low uniform height (500 m - to reduce the image distortion) to generate high-resolution orthomosaics and DSMs of the glacier lake surfaces, empty lake basins, and glacier termini. In
addition to UAV data, we utilized data from KH-9 (1974), ASTER (2000-2019), PALSAR-DEM (June 2008), and SRTM (February 2000) for the computation of lake volumes (Table S2). The SRTM DEM without voids serves as the reference dataset, and the vertical uncertainties of the SRTM DEM are reported to be ±10 m (Rodriguez, Morris and Belz, 2006). The corrected DEMs from the Karakoram region are those used by Bazai, Cui, Carling, Wang, Hassan, Liu, Zhang and Jin (2021) and Gardelle, Berthier, Arnaud and Kaab (2013).

### 3.4. Geometry of lake basin

Although in this study, we have field-derived estimates of lake depth, basin geometry, and lake surface area to calculate lake volumes, in many other applications, only remote sensing data are available for undrained lakes. Consequently, considering that the lake area in satellite images often exhibits a triangular planform (*e.g.*, Fig 2a; 3a), we explored the possibility of using a geometric shape to approximate the volume of undrained lake basins. Such an approach would be valuable where the
depths of lakes are unknown. To this end, we employ NDWI (as in section 3.1) to identify lake outlines through Landsat false-color composites, which use near-infrared, red, and green bands to distinguish water bodies from other features. We employ standard connected component analysis (Dillencourt, Samet and Tamminen, 1992) to manually calculate each lake's area, perimeter, and other surface dimensions (as given in Fig. 3b). Initial calculations are pixel-based and later converted to metric units by multiplying pixel counts with their respective pixel sizes. The pixel size for high-resolution images varied
from 0.8 to 3 m. The output was cross-validated with Khurdopin and Shishper glacier lakes UAV data having a pixel size of 0.063 m and with field survey evidence. Trials demonstrated that the known volume of the lakes determined using DEMs of the lake basins once drained (section 3.3) could be approximated if the length of the lake from the upstream inlet to the ice-dam face ($Z$) and the breadth of the lake at the ice dam ($C$) are known. Given the reported image resolution, uncertainties in the characteristic length measurements (Fig. 3b), measured using GIS 3D interpolation, would translate to uncertainty in lake
volume estimates of only 3% when applying Equations 1 and 2 if lake depth was known exactly. For undrained lakes, assuming the depth is the same as the width of the lake at the dam face (Fig. 3b) likely over-estimates lake volume. It might be expected that geometric estimates based on lake surface area alone would be improved if the lake's depth ($h$) is known at the deepest point close to the dam face. However, in our examples, there is uncertainty in the values of $h$ obtained from DEMs of the drained basins, such that the errors in lake depth estimates translate to errors in lake volume estimates of <14%.
Alternatively, where a lake is present, this latter parameter can be obtained by plumbing the depth from a boat.

Given the triangular shape of the lake surface areas, the first consideration with regard to lake geometry was whether the valley sides might be considered to provide a V-shaped lake cross-section or a rectangular cross-section (Fig. 3b), in either case, regular geometric shapes might provide an estimate of the lake volumes. A rectangular cross-section would be closer to the U-shaped valley cross-sections commonly associated with glaciated valleys. Thus, assuming a V-shaped valley, lake volume ($V$) can be approximated by an irregular tetrahedron (Fig. 3b left-hand panel) where the depth ($h$) is unknown, but the distance from A to B (X in Fig. 3b) and the length $C$ are known values. Assuming the lake surface is an isosceles triangle, and the vertical face at the dam wall is an equilateral triangle, the volume can be obtained from:

$$V = \sqrt{V^2} \tag{1}$$

$$V^2 = \frac{1}{144} [Y_1^2 D^2(Z_2^2 + X^2 + C^2 + E^2 - Y_1^2 - D^2) + Y_2^2 E^2 (Y_1^2 + X^2 + C^2 + D^2 - Y - E^2) + Z^2 C^2 (Y_1^2 + Y + D^2 + E^2 - Z^2 - C^2) - Y_1^2 Y_2^2 C^2 - Y_2^2 Z^2 D^2 - Y_1^2 Z^2 E^2 - C^2 D^2 E^2]$$

where the values for lakeside lengths $Y_1$ and $Y_2$, the main length $Z$, lakeside length $D$, and lakeside length $E$ are defined in Fig. 3b and obtained from geometry.

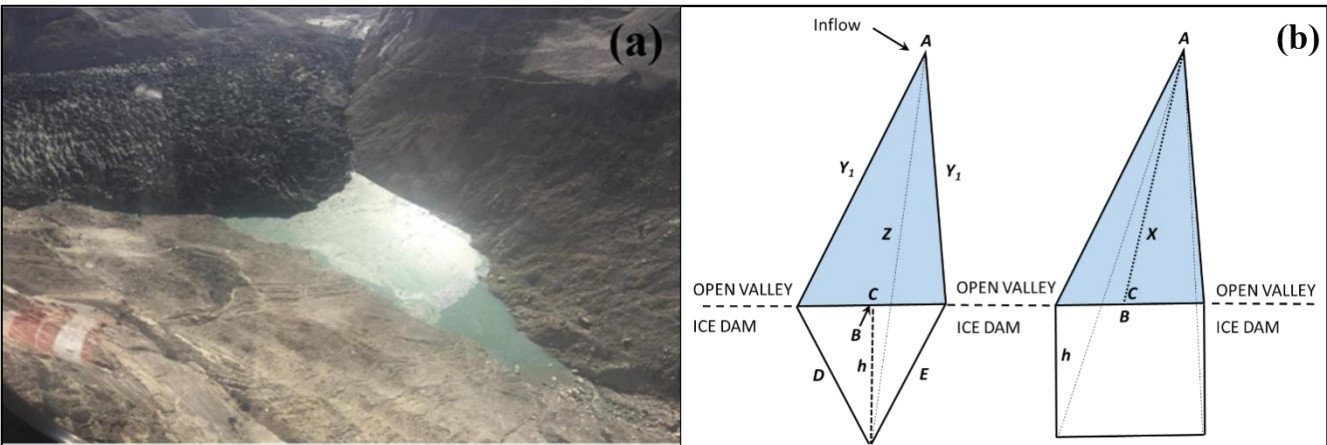

Figure 3: (a) Example of Shishper glacial-dammed lake exhibiting roughly triangular surface 2D shape (see also Fig. 2a); (b) Definition diagram for calculating the volume of the lakes assuming (left) an irregular tetrahedral shape and (right) an irregular pentahedral shape. The blue shading represents the horizontal surface of the lake, and the white area represents the vertical ice wall.

Alternatively, considering a pentahedral, the volume is:

$$V = \frac{1}{3} (h^2)X. \tag{2}$$

These shape assumptions are addressed within the Results.

# 4. Results

## 4.1 Surge velocity and ice dam lake volume

Figure 4a-c presents the relationship between the glacier surge velocity (Khurdopin, Kyager, and Shishper) and 23 GLOFs. The relationship between surge and GLOF was developed using annual average velocity data. The glacier's daily velocity was recorded on the day the GLOF was initiated, as detailed in Table 1. In panels a to c, the GLOF occurred after the peak of the glacier surge, and the resealed lake formed while the surge velocity declined. These responses to slowing the glacier velocity lasted 2-4 years after the surge peak. Thus, GLOFs occur towards the end of a surge period or immediately

afterward; the detail is presented in Table 1. The relationship between the timing of glacier surges and the timing of GLOFs is shown in Fig. 4d, wherein the dates of the GLOFs are given as the month in the year. The three Karakoram glaciers can be used as regional examples of surge behavior controlling GLOF occurrence, as there is a temporal relationship between the occurrence of periods of glacier surging and the occurrence of GLOFs (Fig. 4a-d). This pattern of behaviour prompted the hypothesis that glacier thickening and thinning during surging might control the development of ice-dammed lakes (Bazai,

Cui, Liu, Carling, Wang, Zhang, Li and Hassan, 2022). Lake volumes would increase when the speed of the ice was low, the ice mass would be conserved or increased, and the fracturing of the ice would be reduced. The corollary pertains to when the ice speed increases, the glacier thins, and the fracturing of the ice mass increases, providing hydraulic drainage conduits (Gao, Liang, Qi, Yao, Ma, Mu and Li, 2024). This sequence of events is shown schematically within Fig. 5. A thinning glacier also minimizes potential lake depth and might increase the likelihood of a GLOF occurring over the top of the ice

dam.

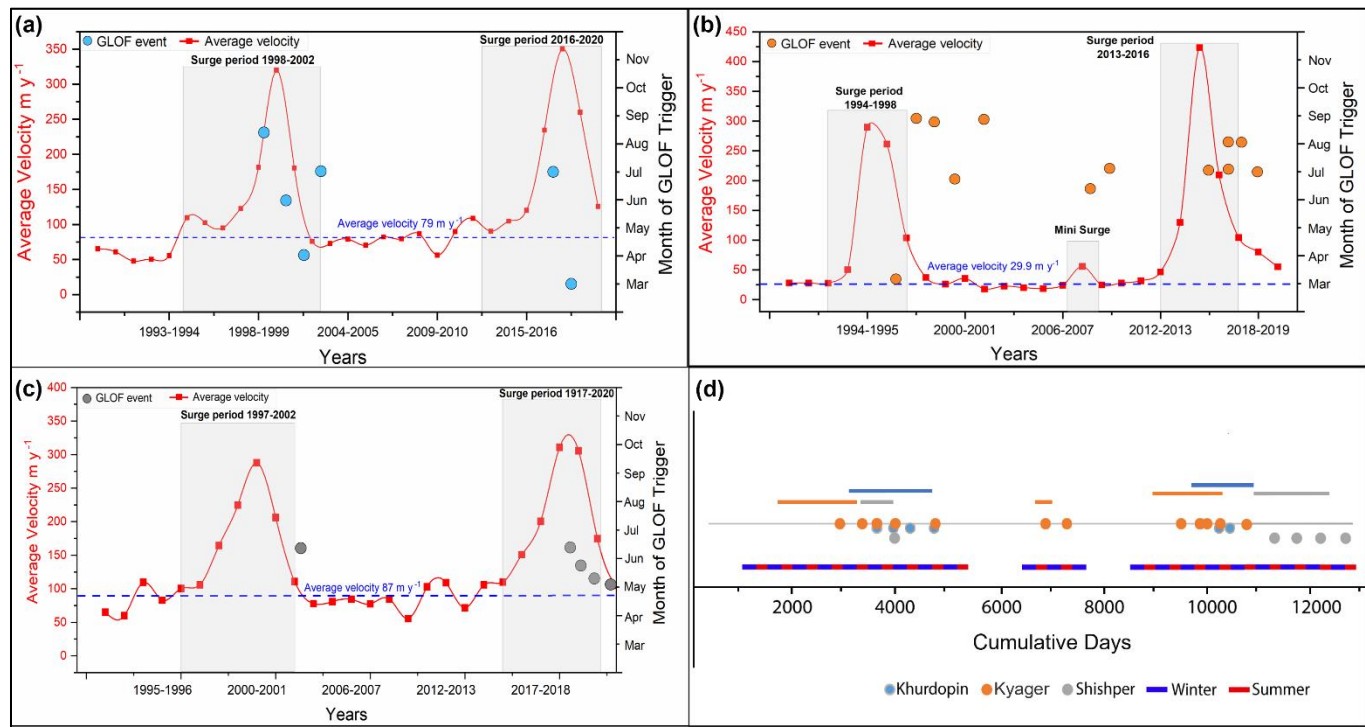

**Figure 4:** Relationship between glacier surges and GLOFs, with average annual glacier velocity during the surge and quiescent phases for three glaciers: (a) Khurdopin, (b) Kyager, and (c) Shishper. GLOFs for these glaciers occurred between the months of March to November. The combined analysis is presented in (d), illustrating the occurrences of GLOFs (dots) and related periods of glacier surging (bars) as cumulative days since 1st January 1990. Some points are plotted below the timeline to avoid coincident positions. The blue and red lines show the winter (from October to April) and summer (May to September) seasons, with the GLOFs occurring dominantly in the summer months. Within panels a to c, the average surge velocity is given as the red curves, and the average velocity during the study period is given in blue text.

As a first attempt to relate glacier behaviour in a predictive sense to lake formation, we sought to determine the relationship between the resulting lake volume from the prior surge speed. Lake volume should be high when the glacier velocity is low, and the ice mass thickens and *vice versa*; there is some support for this assertion (Fig. 6). Within Fig. 6, considering all the data (excluding the three drained lakes), the broad data spread prevents the fitting of a significant least-squares regression function. Nonetheless, trial curve fitting showed that a negative power function would be the best fit.

If the volume of a lake decreases as the glacier surge speed increases, as a negative power function would imply, both the lake depth and surface area decrease; then, from an analogy with a pentahedron, the volume of a pentahedron reduces as the square of the characteristic length (Equation 2), here the water depth. Consequently, assuming the pentahedral analogy applies, a least-squares trendline fitting procedure was used to define the constant, α in the function: $V_{DEM} = \alpha U_s^{-2}$ fitted to all the data excluding the drained volumes. This equation, with α = 13.6, is shown in Fig. 6 and visually is a good fit through data for a range of low values of $U_s$ when lake depths will be greatest. Given the data dispersion and the small sample number, there are no statistical outliers (defined objectively; (Carling, Jonathan and Su, 2022).

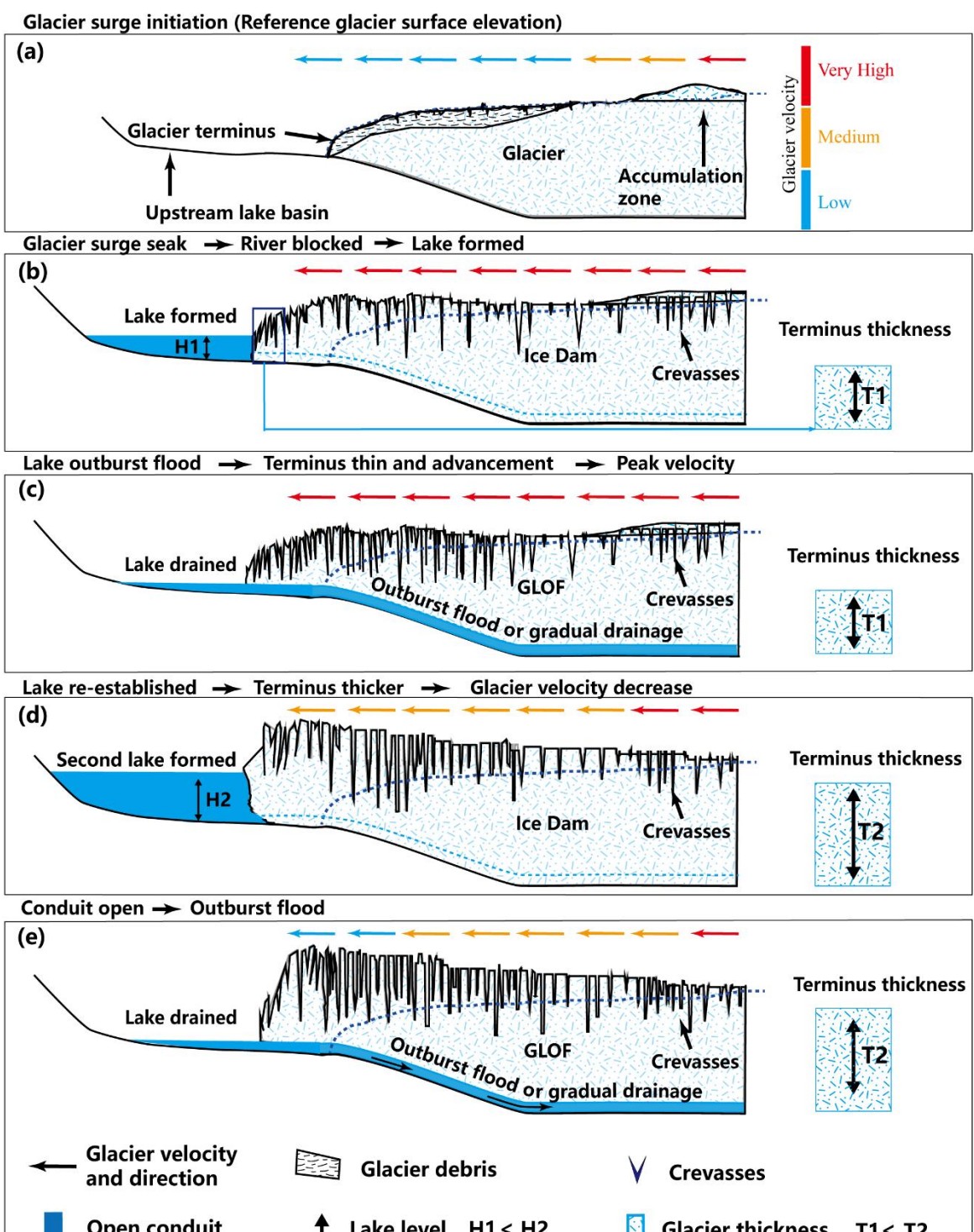

**Glacier surge initiation (Reference glacier surface elevation)**

**(a)**

Glacier terminus

Glacier

Accumulation zone

Upstream lake basin

Glacier velocity

Very High

Medium

Low

**Glacier surge seak** → **River blocked** → **Lake formed**

**(b)**

Lake formed

H1

Ice Dam

Crevasses

Terminus thickness

T1

**Lake outburst flood** → **Terminus thin and advancement** → **Peak velocity**

**(c)**

Lake drained

Outburst flood or gradual drainage

GLOF

Crevasses

Terminus thickness

T1

**Lake re-established** → **Terminus thicker** → **Glacier velocity decrease**

**(d)**

Second lake formed

H2

Ice Dam

Crevasses

Terminus thickness

T2

**Conduit open** → **Outburst flood**

**(e)**

Lake drained

Outburst flood or gradual drainage

GLOF

Crevasses

Terminus thickness

T2

Glacier velocity and direction

Glacier debris

Crevasses

Open conduit

Lake level    H1 < H2

Glacier thickness    T1 < T2

**Figure 5**: The mechanism of glacier surge controls lake formation: a) Prior to the surge, there is no stream blockage in front of a thick terminus; b) in the first year of the surge, stream blockage occurs, leading to lake formation behind a thinning terminus; c) during the first year, peak glacier velocity is reached, the lake drains and the terminus thins; d) as surge velocity decreases the lake reforms and the terminus thickens; e) the second lake can drain as velocity continues to decrease and the terminus thickens.

An eye-fitted power function has been added to Fig. 6 to tentatively define the lower limit of the data spread. These results, although clearly not definitive, indicate that there likely is a relationship between the volume of the lake and the control of the lake water level exerted by the surge speed. Therefore, surge speed should exert some control on lake depth, volume, and potential GLOF volumes.

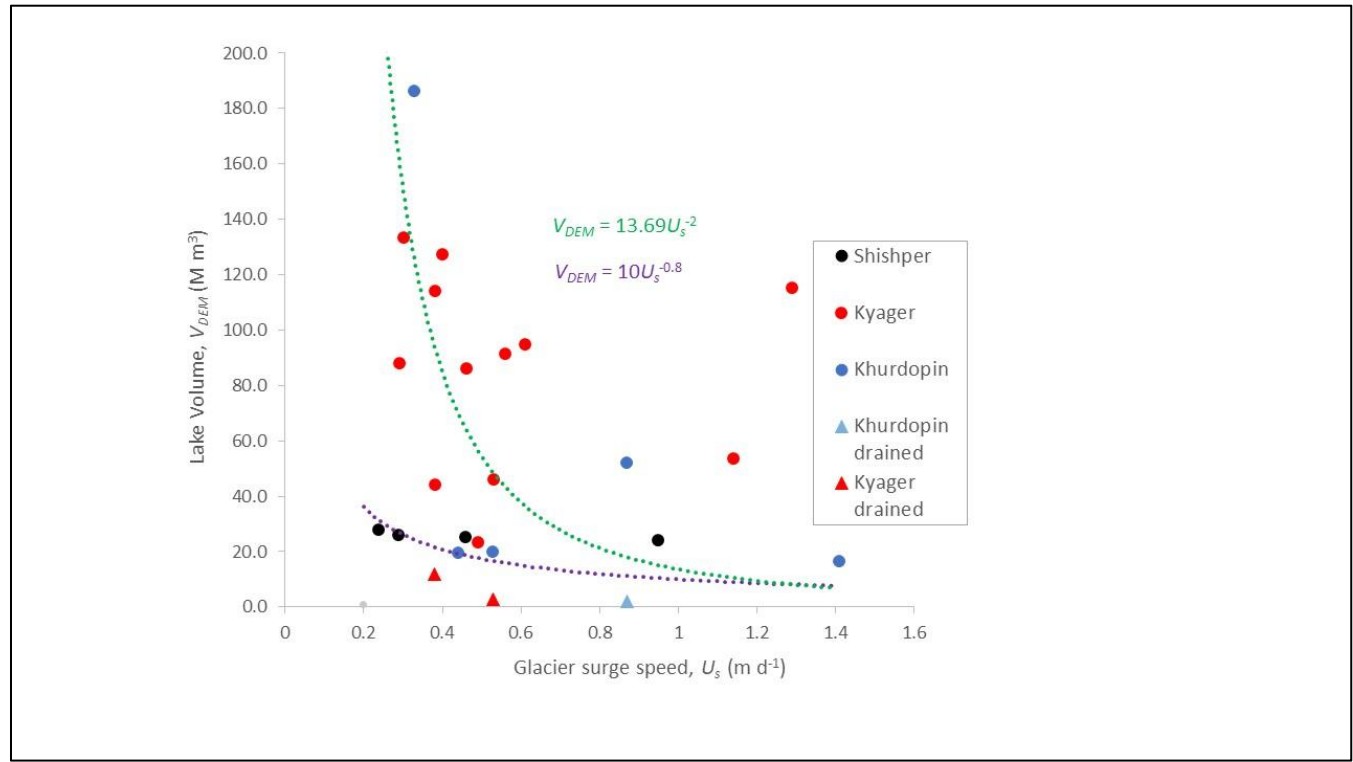

**Figure 6:** Variation in glacial lake volume as a function of the glacier surge speed. Data from three glaciers. Most lakes drained completely, but three drained lakes had residual volumes (triangles). A -0.8-power function (purple curve) defines the lower limit to the data spread, while a -2.0-power least-squares function (green curve) defines the central tendency of the data trend (see text for explanation).

Given the scatter in the data within Fig. 6, additional data would be required to determine if the relationship between surge speed and lake volume does follows a negative power trend, as we have suggested.

### 4.2 Tetrahedron assumption for lake volume

Using Equation 1 and assuming the ice dam face was an equilateral triangle, only the values $X$ and $C$ are required such that the volume of the 'tetrahedron' lakes was around 10 times greater than the volume of the lakes determined using the DEMs (Fig. 7a). This result indicates that the actual depth of the lake ($h$) must be much less than that value associated with an equilateral triangle of side length C (Fig. 3b left-hand side). Nevertheless, this procedure provides a means to estimate lake volume from plan-view data alone.

In contrast to the assumption of an equilateral triangle at the dam face, improved lake volume estimates were obtained considering the measured DEM-derived values of $h$ along with the values of $X$ and $C$. Once again, assuming an irregular tetrahedron as in Fig. 3b, the analysis demonstrated that the tetrahedral lake volume was roughly half that of the DEM volume (not illustrated). This latter result suggests that treating the valley-cross section to be U-shaped (roughly quadrilateral) rather than V-shaped means that doubling the area of the triangular dam face section to form a quadrilateral should provide lake volume estimates, defined as an irregular pentahedron (square-based pyramid) (Fig. 3b right-hand side) closer to the DEM-derived volume estimates.

### 4.3 Pentahedron assumption for lake volume

The pentahedral volume estimates (Eq. 2), as shown in Fig. 7b, are preferable to those values shown in Fig. 7a. They result in a near 1:1 relationship between $V_{Pen}$ and $V_{DEM}$ but require knowledge of the parameter depth: $h$, as well as $X,$ and $C$. If assuming a rectangular base to the pentahedron provided an exact match to the DEM volume, the correlation coefficient value would be unity. Thus, the coefficient of 0.88 reflects the deviation of the cross-sectional shape of the lake at the dam face from a rectangle. Note that the relationships between both determinations of lake volume (Fig. 7 a and b) progressively deviate from a 1:1 relationship as lake volume increases. This trend might indicate that larger lakes are less well-defined as tetrahedrons or pentahedrons as the volumes increase.

Assuming a tetrahedral shape to a lake, then the lake volume can be estimated from remote sensing images alone as only the length of the lake ($X$) and the breadth of the lake ($C$) at the ice dam are needed to estimate the lake volume. Assuming a pentahedral shape, the depth ($h$) of the lake at the ice dam is required as well. Although subsequent to GLOF drainage, $h$ can be measured from a DEM or field survey, for the purposes of mitigation, a warning prior to a GLOF occurring is preferable.

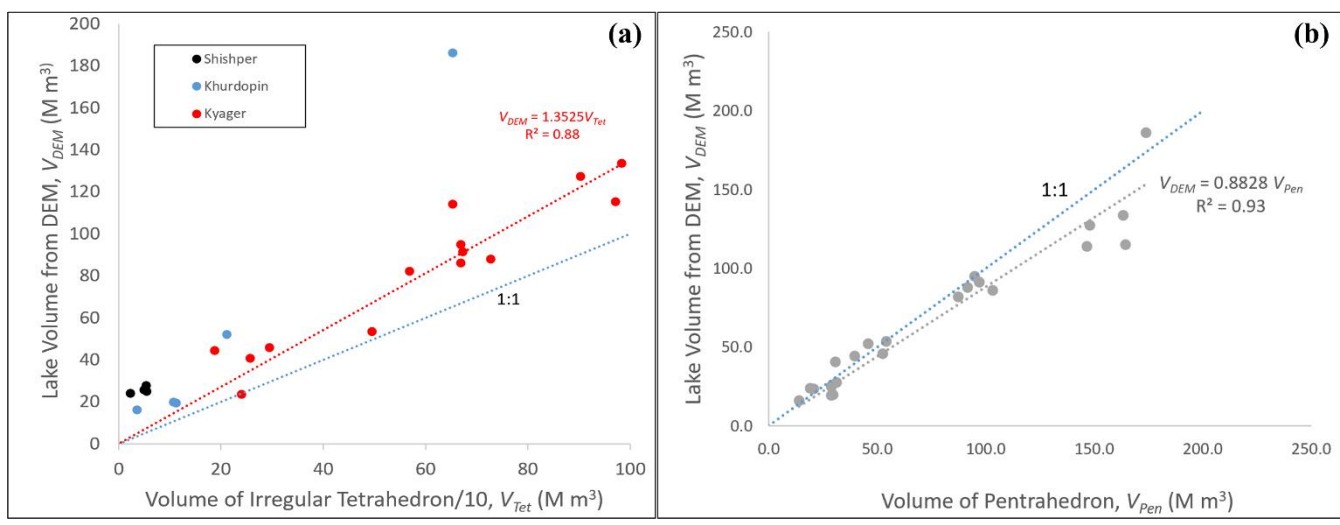

**Figure 7:** Panel **(a)** Relationship between the volumes of irregular tetrahedrons/10, derived from Eq. 1, and the volumes of the lakes determined using DEMs. Panel (b) The relationship between the volumes of irregular pentahedrons and the volumes of the lakes was determined using DEMs.

### 4.4. Anticipating the Timing of GLOF Events

The timing of a GLOF remains difficult to determine, but for tunnel drainage, the main driver is the critical depth (Thorarinsson, 1969). The critical depth is important as it is the depth that exerts sufficient pressure at the ice dam wall to induce completed connectivity within the sub-glacial GLOF drainage conduit (Gao, Liang, Qi, Yao, Ma, Mu and Li, 2024; Yasuda and Furuya, 2013). For the cases of Shishper, Khurdopin, and Kyager, the glacial lake depths ($h$) have been normalized by dividing by the minimum value of the ice dam height to give values ($n'$), normalized lake depths, that range between 0 for a fully-drained lake to a hypothetic value of 1 if the lake level reached the height of the ice barrier. At the approximate time of GLOF occurrence, the resulting values of $n'$ range between 0.32 and 0.95 (Fig. 8a-c). Most GLOFs occur for a range of $n'$ values between 0.61 and 0.95 (Fig. 8d). Thus, $n' = 0.60$ can be regarded as a warning level value with the potential for a GLOF occurring imminently increasing as $n'$ approaches unity.

As the water pressure ($P$) at the dam face increases linearly with water depth in each lake, any variation in the pressure with $n'$ that deviates from the linear trend reflects changes in the height of the ice dam (Fig. 8d). Thus, for example, the values of pressure for the Shishper lakes around an $n'$ value of 0.8 reflect greater overall deeper and higher ice dam heights in contrast to the Kyager lakes for similar values of $n'$. The two values of low pressure for $n' < 0.6$ are associated with relatively low ice dams and consequently reflect presumed low structural integrity within the ice mass, allowing ready conduit development. In the present examples, low values of $n'$ ($< 0.6$) likely are associated with shallow lakes of low hazard potential. Overall, the Kyager data (Fig. 8d) indicate that a minimum water pressure of around 500 kPa should be regarded as a threshold for general concern for GLOF occurrence in the region. However, consideration should be given to local conditions when applying the findings of this study to other locations around the globe.

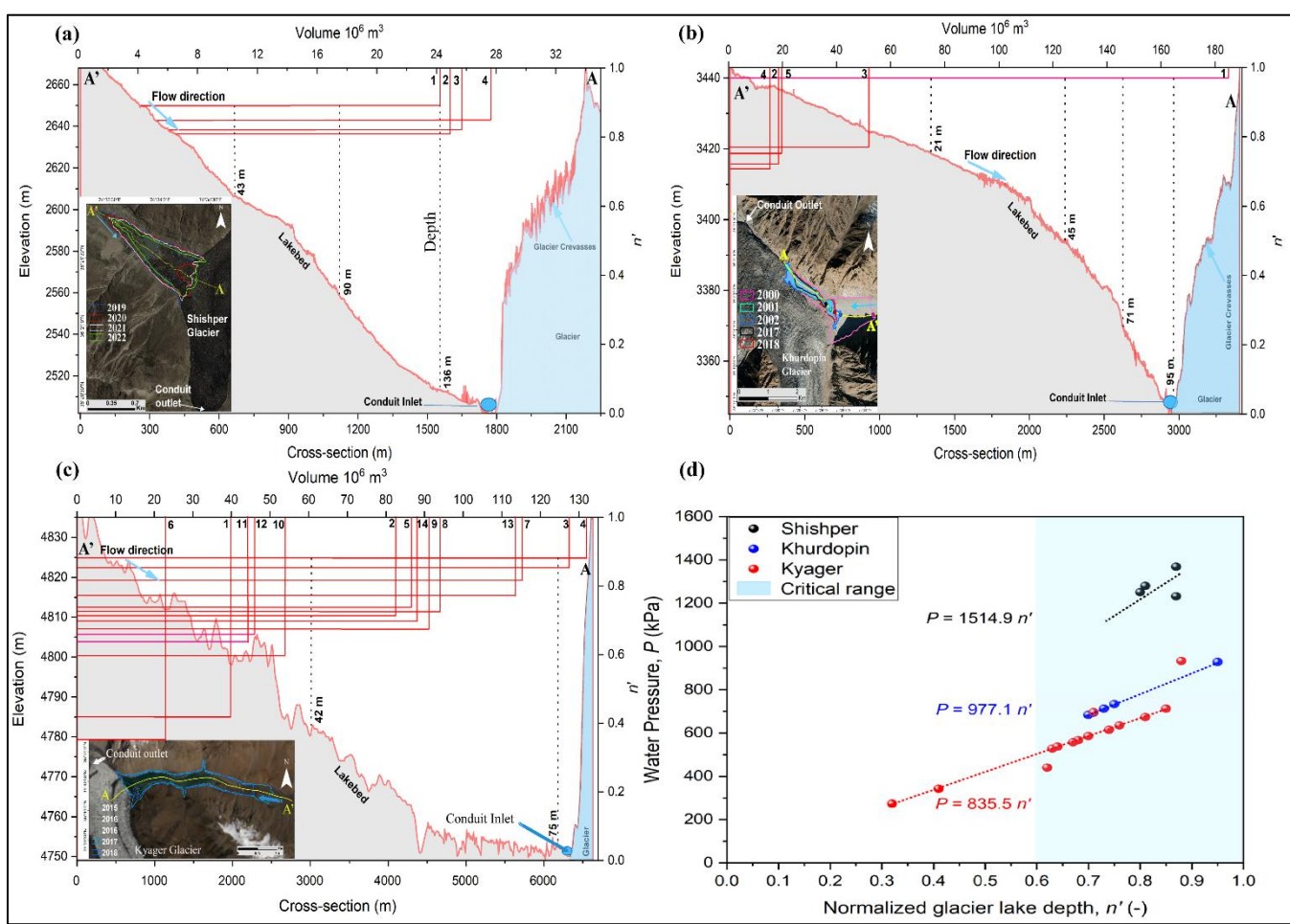

**Figure 8:** The relationship between lake volume and elevation and the critical normalized lake depths for GLOFs. a) Shishper; b) Khurdopin; c) Kyager, redlines connect the relevant elevation and volume in each instance; d) Water pressure at dam face as a function of $n'$. The cross sections are denoted by (A' and A). The serial number and date of each lake flood event are shown in Table 1. The straight, solid red lines relate specific lake elevations to volumes. An UAV photograph captured in the field was used for panel a, the images captured by the GF-2 were used for panel b, and Landsat OLI 8 was used for panel c.

## 5. Discussion

Although predictive models related to ice-dam lake development and subsequent GLOF hazard would best be based on modelling the physics of the systems, the controlling parameters are numerous and complex. For example, the mechanisms of glacial sliding, over-burden pressure, tensile and driving stresses require consideration, as do flexure and ice fracture mechanics, thermal erosion, and water pressure, amongst other controls (Carrivick et al., 2020; Joseph et al., 1996) along with climatic influences (Ng, 2007; Richardson and Reynolds, 2000). Few of these controls are well-understood, and, importantly, even where there is an adequate theory, the field data required to inform modelling are absent for specific

potential GLOF locations. In this study, the glacier and lake interactions and their empirical relationships have been explored, and their effect on lake volume and draining processes examined. Understanding glacier surges, lake formation, and the interactions between lakes and glaciers is crucial for advancing knowledge and developing empirical or numerical GLOF models in mountainous regions (Carrivick, Tweed, Sutherland and Mallalieu, 2020; Quincey and Luckman, 2014). Glacier surge speed is routinely determined using remote-sensing imagery (Paul, 2015), as is lake surface area (Quincey and Luckman, 2014). Thus, remote sensing provides a means to develop images similar to Fig. 6 for specific locations around the globe where ice-dammed lakes form due to glacier surging. Although the data within Fig. 6 are scattered, a negative relationship between surge velocity and lake volume is strongly implied. Specifically, data points scatter around a median trend according to a theoretical -2 power function (Fig. 6). Clearly, more data points within Fig. 6 would be desirable such that the relationship (if any) between ice surge velocity and lake volume might be better defined.

There is an urgent need for simpler methods to predict the probable triggering water levels that lead to GLOFs and the likely volume of the ice-dammed lakes that translate into GLOF hydrographs. Given that requirement, it is acknowledged that the relationships proposed herein are empirical and apply specifically to glaciers within the Karakoram region. However, there is no reason to suppose that similar functions based on geometric considerations (Zhang, Bolch, Yao, Rounce, Chen, Veh, King, Allen, Wang and Wang, 2023) and a critical depth (Zhao, Yang, Walling, Zhang and Zhang, 2017) might not be developed elsewhere, including for moraine-dammed lakes (Yao, Liu and Wei, 2010). Below, the approach is explored for glacial ice-dammed lakes worldwide.

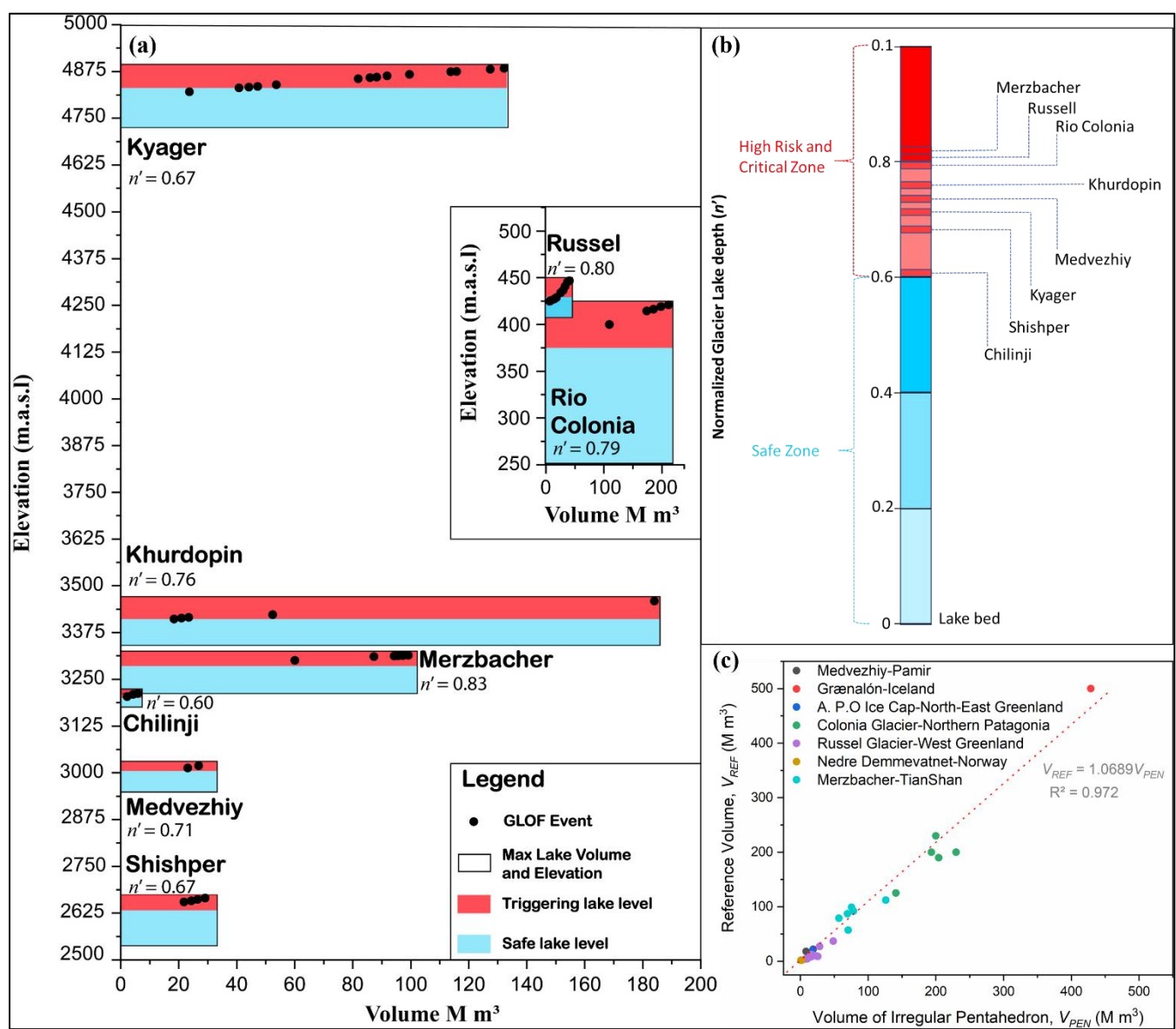

**Figure 9: (a)** The maximum volume of ice-dammed glacier lakes that result in GLOFs related to water surface elevation. In this representation, the red bands are defined by the range of $n'$ values recorded just before each GLOF event was initiated. The blue band represents lakes with volumes below the normalized lake depth of 0.60. **(b)** Normalized critical lake depths and the high-risk and critical zones for GLOFs, with an inferred gradation for risk within the safe zone, are shown by shades of blue. **(c)** The relationship between the reference volume ($V_{REF}$) and the measured volume ($V_{PEN}$) was obtained through a geometric approach. The location of these glaciers is presented in Fig. S1.

Despite the absence of long-term records, those that are available indicate that glacial ice-dammed lakes worldwide exhibit consistent behavior in terms of lake formation, filling, and volume gain in response to low glacier velocity (Bazai, Cui,

Carling, Wang, Hassan, Liu, Zhang and Jin, 2021). Additionally, specific water pressure and critical normalized lake depth values for initiating outburst floods are evident (Fig. 8).

Building upon the information presented in Figure 8 for the Karakoram, Figure 9, based on 50 GLOF events from ten glacial ice-dammed lakes, offers a depiction of the conditions under which glacier lake volume measurements are estimated with high accuracy. Considering lakes other than within the Karakoram, the lake elevation (Fig. 9a) at the time of a GLOF was available for five lakes that have triggered more than twice (Chilinji, Medvezhiy, Merzbacher, Russell, Rio Colonia),

enabling the calculation of the normalized lake depths (Fig. 9b). The data points in Fig. 9c represent 27 lakes within the Pamir, Tianshan, Greenland, and northern Patagonia, for which lake volume data are available in the literature. These latter data were used to establish a relationship between the lake volume estimated using Equation 2 and the reported volumes ($R^2$ = 0.972; Fig. 9c).

Figures 9a and b serve as key components and summaries for this discussion, identifying the lake volumes, elevations, and

465 the critical normalized depth values for GLOF outbursts. Critical normalized lake depth values (*n'*) exceed 0.60 in all cases of GLOFs (Fig. 9b). From this result, we infer that a safe lake level can be defined as < 0.60, while the trigger level is ≥ 0.60. Values of *n'* < 0.60 were associated with slow, non-catastrophic lake drainage. Therefore, in the case of future ice-dammed lakes, values of critical depth (*n'*) exceeding 0.60 should be a cause for concern, and 0.60 would serve as a warning level.

Estimating the volume of an undrained ice-dammed lake from a field survey is dangerous due to floating ice, rugged terrain,

and sudden drawdowns. The utilization of DEM measurements for lake volume estimation may also introduce high uncertainties or errors due to the difficulty in defining the lake depths (Carrivick, Tweed, Sutherland and Mallalieu, 2020; Emmer, 2018). However, for rapid response or mitigation policy purposes, the empirical model (Equations 1 or 2) used in the current study proves to be quite efficient when applied to estimate the lake volume before a GLOF, not least because the errors in measurements from both satellite and UAV images are now quite small, as noted within the Method.

Although GLOFs cannot be predicted from this approach, the likely volume of water that might be released catastrophically can be determined. For sites that are deemed to pose a threat to human life and infrastructure, once the lake volume is better constrained, either through DEM analysis or geometric considerations, the value of *n'* for any specific lake provides a ready indicator of the probability of an imminent GLOF. In contrast to the lower trend for water pressures associated with the Kyager lakes, the higher water pressures required to cause the Khurdopin and Shishper lakes to empty may reflect greater

structural integrity, possibly related to a greater downstream extent of the glacier dams. These structural issues can be examined in the future. Still, at this stage, if *n'* exceeds 0.6, an initial general warning could be issued to communities downstream of the ice dam. In principle, the estimated volume of a potential GLOF can then be routed downstream using standard hydrodynamic flood routing procedures to determine the timing, depth, and extent of flooding at locations where inundation is forecast. Thus, the severity of the likely impact on humankind can be determined, and specific warning times

can be derived from the modelled rate of travel of the GLOFs. These results represent a step forward from the observations made by Carrivick, Tweed, Sutherland and Mallalieu (2020), who proposed the exploration of the interaction between lake

water and glaciers to understand the lake formation process and identify lake depth, level, and volume. Based on this understanding, empirical models can be generated to predict GLOFs in a timely manner.

## 6. Conclusion

Despite an escalating hazard from glacier lake outburst floods (GLOFs), understanding these hazards remains limited. It is imperative to determine the causes of these hazards, make timely predictions, and formulate new mitigation policies to minimize losses. Herein, it has been shown tentatively that glacier surge speed may correlate negatively with ice-dammed lake volumes such that glacier dynamics control lake volumes. Consequently, in those cases where glacier surge speed is monitored, trends in surge speed can provide a timely indication that a lake might form, allowing the risk of a potential

GLOF to be considered and mitigation measures reviewed. Identifying the critical depths, lake volumes, and pressures of ice-dammed lakes worldwide associated with GLOFs has indicated that GLOFs may be imminent when the normalized depth ($n'$) for lakes exceeds a critical value ($n' = 0.60$) with a typical water pressure on the dam face exceeding 510 kPa. Identifying a critical depth that might lead to a GLOF is relatively straightforward and, thus, is a useful measure that provides a timely warning for downstream communities. Comparing published surveyed lake volumes with geometric

volume estimates for 23 GLOF events from the Karakoram and 27 events from around the world, linear least-squares regression ($R^2 = 0.972$) demonstrated that geometric estimates can be robust in the absence of detailed field or remote-sensing surveys. Such an approach to determining lake volumes is useful on two accounts. Firstly, the approach can provide a quick estimation of lake volume. Then, if the lake volume is considered to be of concern more detailed survey work can be commissioned to obtain a more accurate estimate of the volume. Secondly, in those situations where sufficient resource may

not be available to conduct detailed volumetric surveys the geometric approach provides a ready tool to obtain a reasonable volume estimate. Taken together, these findings suggest that future exploration should concentrate on specific volume and depth parameters to determine critical thresholds associated with normalized depth and the associated lake volume for future predictive purposes. In this respect, it should be noted that the water pressure recommened herein as potentially of concern (510 kPA) pertains to lake depths of c. 50 m, whereas deeper lakes in other regions might drain at different values of

pressure.

### Data availability

Data and/or further information regarding the data used can be obtained from the author(s) upon request.

### Supplementary Information

Supplementary information for this paper is available in supplementary material and at https://doi.org/10.1016/j.earscirev.2020.103432 and https://doi.org/10.1016/j.gloplacha.2021.103710.

**Author Contributions**

N.A.B. conceptualized and designed the study and methodology, generated and compiled field data, processed data visualization, and drafted the manuscript. P.A.C. contributed to conceptualization, methodology, interpretation, discussion, review, and editing. P.C. supervised funding acquisition and commented on the paper, and H.W. contributed to compiling the field data. N.A.B., G.T.Z., J.W, D.Z.L, and J.H contributed to remote sensing data analysis.

**Competing interests**

The authors declare no conflict of interest.

**Disclaimer.**

Publisher's note: Copernicus Publications remains neutral with regard to jurisdictional claims made in the text, published maps, institutional affiliations, or any other geographical representation in this paper. While Copernicus Publications makes every effort to include appropriate place names, the final responsibility lies with the authors.

**Acknowledgments**

Special thanks to the monitoring team of the Gilgit-Baltistan Disaster Management Authority (GBDMA), Quaid-i-Azam University, and Karakoram International University for their support, data sharing, and technical assistance. We are also grateful to Iqtidar Hussain for sharing his expertise and to the Special Research Assistant Program of the Chinese Academy of Sciences. Additionally, we deeply thank the editor and the three reviewers for their valuable and constructive comments.

**Financial support**

This study was supported by the National Natural Science Foundation of China (Grant No. 42350410445) and the Second Tibetan Plateau Scientific Expedition and Research Program (STEP) (Grant No. 2019QZKK0906).

**Review statement.**

This paper was edited by Tobias Sauter and reviewed by Furian Wilhelm and two anonymous referees.

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
