# Peer review of "Refining Lake Volume Estimation and Critical Depth Identification for Enhanced GLOF Event Anticipation"

_EGUsphere, 2024_

## Referee Comment (RC3)

[referee-annotated manuscript omitted]

---

## Author Comment (AC2)

**First Referees comments**

**Review: Predicting the Risk of Glacial Lake Outburst Floods in Karakorum**

The authors of the article "Predicting the Risk of Glacial Lake Outburst Floods in Karakorum" turn to a very important topic, the investigation of the trigger mechanisms of GLOFs. The article contains interesting theories and examines intriguing relations in this context. The article is well written and for the most part logically structured. However, the flow of reading is sometimes impeded because paragraphs do not follow on logically from one another, especially in the methodological chapters 2.2-2.4. But there are also some backtracks later on when methods and contexts are explained instead of just presenting the results of the scientific work.

Thank you for noting that the manuscript is well written and indicating where some restructuring would benefit the flow of the argument.

In particular, the focus on determining the lake volume appears somewhat convoluted and, in some cases, does not refer enough to previous research. Hypotheses are put forward and rejected; it is not always clear what the added value is compared to previous methods (e.g. Cook and Quincy (2015) or the paper by Zhang et al. (2023) already mentioned in the comments). Chapter 3.1 on the relationship between water depth and GLOF event is more interesting and also more fluently written, to my mind.

Thank you for your valuable feedback. We have made significant revisions to address your concerns about the convoluted nature of the lake volume determination section and the need for better contextualization with previous research. The methodology section now provides a clearer, more logically organized account of our approach. We have added explicit references to Cook and Quincy (2015) and Zhang et al. (2023) to better situate our work within existing research and highlighted how our integration of high-resolution UAV data with historical remote sensing datasets offers improved accuracy in lake volume estimations, reducing uncertainties and enhancing GLOF event predictions.
The revised results section 3.1, now 4.1, titled "Surge Velocity and Ice Dam Volume," presents relationships and hypotheses more explicitly, with improved data presentation and detailed explanations to support our findings. This section maintains the fluency and clarity of Chapter 4.1, ensuring consistency throughout the manuscript. We believe these revisions address your concerns and significantly enhance the clarity, coherence, and scientific contribution of our work. Thank you again for your insightful comments.

Although the Discussion contains some parts that would have been better placed earlier in the manuscript as they contain new sources and information, it is well written and presents the results of the study, its limitations and achievements well and puts them in context with previous publications.

Thank you for your positive feedback on the Discussion section. Based on your suggestion, we have revised and updated it to ensure that all new sources and information are appropriately

integrated into earlier sections of the manuscript. This restructuring enhances the overall fluency and logical flow of the manuscript, making the narrative more cohesive. We are confident that these changes have improved the clarity and readability of our study. Thank you again for your valuable insights.

The illustrations support the article in the right places, but could benefit from a revision of the color schemes and, in some cases, a little more scientific clarity in the statements.

Illustrations have been resided in line with your suggestions.

Overall, the results are interesting and contribute to the current state of science, especially with regard to the relationship between lake depth, water pressure and the timing of a GLOF. With the editing of some minor revisions, I see no problems for publication and congratulate the authors on this work.

Thank you for your encouraging feedback and positive assessment of our work. We are pleased to hear that you find our results interesting and a valuable contribution to the current state of science, particularly regarding the relationship between lake depth, water pressure, and the timing of GLOF events. We appreciate your recognition of our efforts and are committed to addressing the minor revisions you have noted. We are confident that these final adjustments will further enhance the quality of our manuscript. Thank you once again for your constructive comments and support.

References:

Cook, S. J. and Quincey, D. J.: Estimating the volume of Alpine glacial lakes, Earth Surf. Dynam., 3, 559–575, https://doi.org/10.5194/esurf-3-559-2015, 2015.

Zhang, T., Wang, W., and An, B.: A conceptual model for glacial lake bathymetric distribution, The Cryosphere, 17, 5137–5154, https://doi.org/10.5194/tc-17-5137-2023, 2023.

**Specific comments:**

At the beginning of chapter 2, the authors mention three glaciers. At this point, the reader wonders: Why were these glaciers and their lakes selected? It would facilitate the flow of reading if the authors could explain this a bit at this point. Later, in P9L232, the authors mention, that "the three Karakorum [sic!] glaciers can be used as regional exemplars [sic!] of surge behavior controlling GLOF occurrence". This is too late, this needs to be explained much earlier. Maybe the authors could add a short subchapter about the study area, indicating its location in HMA and the position of the individual glaciers?

An additional chapter (2) has been added to explain the context of choosing the three glaciers for investigation. The new Figure 1 provides an overview of the location of the three glaciers within High Mountain Asia.
**"Study Area**

The Karakoram Mountain ranges in HMA are known for their complex geology, climatic variability, and denudation processes, including debris flows, mudflows, landslides, rockfalls, avalanches, and GLOFs. As was noted in the preceding section, changes in glacier dynamics, increasing glacier surges, and a trend of increases in GLOF-related disasters characterize this region. These hazards are responsible for substantial economic losses, including the destruction of residences, infrastructure such as roads and bridges, and agricultural areas, as well as blockages of transportation routes like the Karakoram Highway and other expressways (Shrestha et al., 2023). Glacier surges in the region have been recorded since the 15th century (Bazai et al., 2021). Since the application of remote sensing to the monitoring of the glaciers from 1970 to 2020, an increasing occurrence of glacier surges has been recorded from the 1990s, with some glacier surges being linked to the formation of ice-dammed lakes and subsequent lake outburst floods. Some lakes persist only seasonally, forming in the winter when temperatures are very low and draining slowly in the spring or summer. Other lakes are more persistent (Bhambri et al., 2019; Hewitt and Liu, 2010), and pose the potential for catastrophic outbursts. The most frequent glacier surges and formation of lakes leading to outburst floods in the Karakoram region occur for the Khurdopin, Kyager, and Shishper glaciers.  Although the foreland of the Kyager Glacier, situated in the Shaksgam Valley, is uninhabited, GLOFs have caused damage and losses further downstream. Conversely, GLOFs from the Khurdopin and Shishper glaciers, located in the densely populated Hunza area, have resulted in casualties and substantial economic losses. Consequently, these glaciers and their lakes are selected for study.  The focus of the broader investigation is to obtain the data necessary to understand the complex behaviour of the glaciers and their drainage systems with a view to anticipating when the occurrence of GLOFs is imminent. Thus, there is an urgent need to identify trigger factors for GLOFs to provide downstream warnings in a timely fashion.  A better understanding of the complex process behaviours should eventually lead to improved prediction of such events, not only within the Karakoram but also worldwide.

[Figure]

Figure 1: Overview of the study site in the Karakoram (a) and the High Mountain Asia (HMA) region; panels (b-d) present the extent of each glacier at a given time that surge speed has led to ice-dammed lake formation. The associated ice flow velocities are indicated. The background of panels (b-d) comprises, and the source is © Google Earth mages.

Figure 1 gives a good overview on the two lakes, but some things could be improved. Unless I missed a meaning of the color of the year lines, I would suggest using a fixed color scheme for the specific years, so that each year has the same color in each image. What is symbolized by the squiggly blue arrows?

Figure 1 in the original manuscript is now Fig. 2. We have modified the figure and the caption to address the referee's points.

Chapters 2.2 to 2.4 are very important as they explain the scientific basis of this article. However, all three chapters blur into another as it is not always easy to follow the structure of the article. For example, in 2.2, the eponymous "glacier mapping" is explained with the use of remote sensing and field observations and the lake volume calculation is introduced. Then, chapter 2.3 is called "field observation and lake volume measurement" – even if both have been mentioned before and even though the volume calculation especially is the focus of 2.4. A bit of restructuring and more suitable subtitles would improve the flow of reading.

The text within these two sub-chapters has been revised to make the sections distinct.

At the end of 2.3 (P8L189), you mention another way to compute lake volume – from DEMs. What method did you use here?

Lake volumes were calculated using the DEM of the empty lake basin from the mentioned DEM data or UAV data closest to the GLOF events, together with the lake extent and lake surface altitude obtained from high-resolution images. Field surveys of the Shishper glacier lakes were conducted in 2019, 2021, and 2022, and for Khurdopin in 2017 and 2018, using hand-held GPS and UAV (see section 3.2) to determine annual lake extents, lake depths, glacier heights and thickness, termini positions, and glacier surface displacements. The same process was applied to all ice-dammed lake outburst sites, and KH-9, ASTER, PALSAR-DEM, and SRTM were used for Kyager lakes.

It is difficult for me to understand the reasoning behind chapter 2.4. At the moment, it appears to be a mix between presenting the methodology and results, which does not improve the flow of reading. In the previous chapters you have explained the method of estimating/calculating volume (e.g. using field data, remote sensing, DEMs) – and now the authors present another chapter describing yet another method of estimating the lake volume using the lake basin approximation, even though they have previously stated that they have access to the lake basin once the lake is drained, by plumbing or with the help of DEMs. It would help the reader if the reason for this chapter was made clearer.

The 2.4 section has been updated to 3.4, and Text has been added to the beginning to explain the reasoning. Often, only remote sensing data are available, making it difficult to estimate the lake volume. This section introduces the approach of using simple geometric shapes to define lake volume.

**As for the content of this subchapter:**

Maybe I missed it, but Fig.2a contains the pentahedral lake shape, but only shows the comparison between DEM-based volume and tetrahedral volume in Fig2b. What is the reason for the pentahedral diagram?

The pentahedral assumption works better than the tetrahedral assumption. This issue is now explained in the revised text both in the methodology and result section.

Finally, what was the reason for assuming that the vertical cross-section of the lake at the dam is an equilateral triangle? Is this a common geometry for this kind of lakes? Also, if I understand correctly, this approach does not provide useful data because the estimated volumes are far too high (Fig2b). The authors state that lake depth is probably much lower, but do not present alternate estimations. Now of course it is completely acceptably to include an unsuccessful approach, but it should be (a) made clearer to the reader or (b) maybe moved to the Results or Supplements section.

The assumption of an equilateral triangle at the dam face is an inevitable outcome of the consideration as to whether the lake body is best described as an irregular tetrahedron. The alternative is a regular rectangle. In this section, we are developing a Method to explore whether this is the form of natural lake basins. The approach does provide a useful estimate in that the

misfit between the calculated volume and the observed volumes is systematic, allowing a potential correction to be applied to volumes calculated from remote sensing data alone. We have indicated at the end of this section that the implications of these calculations are considered within the Results section.

Figure 3 shows the very interesting correlation between surge periods and the occurrence of GLOF. It is a valuable contribution to this article. Unfortunately, it takes too much time to fully understand it. I would encourage the authors to spend more time on making this figure more precise and consistent, as this kind of figure is so important for the reader to fully grasp your work.

The figure has been updated accordingly to make it clearer and easier to read. In the revised text, it is now Figure 4.

[Figure]

Figure 4: Relationship between glacier surges and GLOFs, with average annual glacier velocity during the surge and quiescent phases for three glaciers: (a) Khurdopin, (b) Kyager, and (c) Shishper. GLOFs for these glaciers occurred between the months of March to November. The combined analysis is presented in (d), illustrating the occurrences of GLOFs (dots) concerning periods of glacier surging (bars) as cumulative days since 1st January 1990. Some points are plotted below the timeline to avoid coincident positions. The blue and red lines show the winter (from October to April) and summer (May to September) seasons, with the GLOFs occurring dominantly in the summer months. Within panels a to c, the average surge velocity is given in red text, and the average velocity during the study period is given in blue text.

One example: the labeling of the x-axis is very strange in 3a-c, as the same interval sometimes means one year and sometimes three years. The vertical labeling is also difficult to read. A different type of labeling that, for example, only shows every five years (but written horizontally) and has constant axis ticks for the years between the labels would be much more appropriate and would clean up the picture a lot.

Modified and updated

Also, the font sizes are not consistent. Fig3b and Fig3c are missing a legend (comparable to Fig3a) which would explain the differently colored dots.

Modified and updated

And even after some consideration, I am not sure I understand Fig3d. The date seems to be 01/01/1990, but also some kind of timeline is indicated by the rising numbers below the bars. If this panel is only a combination of 3a-c, the authors should explain it more.

Modified and updated

In Fig. 4 and the preceding paragraph, the authors mention the exclusion of outliers from their data. In Fig.4, only two of the four excluded outliers are shown, but at least one of them appears to be relatively close to the main data spread. I would encourage the authors to include the reasoning behind the detection and treatment of the outliers – e.g., are they errors in the measurements, are they not representative etc.

Thank you for these observations, which we respond to below.

In general, Fig.4 is a valid support for the interesting investigation in the connection between surge velocity and lake size. However, it could be improved by a clearer legend and a more distinctive color mapping. At the moment, one color (light blue) is used for a trendline, glacier points, and the background shading (which should have a legend entry). Also, the triangles are explained two contradictory ways: In the caption, the authors describe the triangle lakes to have residual volume. In the legend, the triangles indicate drained lakes.

Thank you for these suggestions. The original Figure 4 is now Figure 5. These queries prompted us to reconsider the detection of outliers, so we sought the advice of a professional statistician. There are no statistical outliers, so we have redone the regression analysis and simplified the diagram. The interpretation of these results has not changed but should now be better expressed. The color scheme has been altered, along with caption and text modifications. The shading has been removed.

In Chapter 3 (P11-12 from L265-L290), the authors repeat the contents of chapter 2.4. They again mention the theory of the lake cross section resembling either an equilateral three-sided triangle or a square at the dam face. However, this approach was already described as unsuccessful in 2.4. due to the resulting volume estimations being much too high. As this section explains some methods more concise than before, these parts should be merged with its already existing counterparts in the Methodology, while the parts of the Methodology in 2.4 dealing with actual results should be moved here.

The Methods and Results chapters have been substantially revised to eliminate repetition and increase the clarity of the argument.

Figure 6 appears very full, but it conveys interesting results, especially in 6d. However, I would encourage the authors to spend some more time to "clean up" the figure. Some fonts are extremely small – maybe some information can be put into a separate table (like the L, E, and V data between c and d). I also wonder why there is no inset card for 6c.

Modified and updated

**Technical corrections:**

P1L1ff: The mountain range is called Karakoram, to distinguish it from the city of Karakorum, if I'm not mistaken. Please change it throughout the text.
Done

P2L40: Why is "Moraine Lake" capitalized?
Corrected

P3L92: "The current research focuses of Bazai and colleagues aims to leverage…" The plural of focus should be foci. à "The current research foci/priorities of Bazai and colleagues aim to leverage…"
Corrected

P3L98: A very nitpicky comment: In this sentence, the authors claim to have used "all available open and commercial satellite imagery sources", and they list a lot of sources, but surely not all. Please clarify if you indeed used the others, or rephrase the sentence.
Corrected

P4L124: Maybe a translation issue: The authors explain that lake extent and surface level can be used to "measure" the lake's volume. To my mind, "estimate" or "approximate" would be more accurate, because to really measure a lake's depth one would need to study its bathymetry.
Corrected to 'estimate'

P5L132: Probably will be fixed during editing, but "Gilgit-Baltistan Disaster Management Authority" has a smaller font size than the rest.
Corrected

P5L136: I would suggest swapping this paragraph with the next paragraph. With a little rewording, the reader would then first learn that all three glaciers are surging glaciers, and then learn about the method of measuring velocity. See also specific comment 1.
Modified and updated

P6L156: There is a colon in this line, the meaning of which I do not understand.
Text modified

P6L157: Maybe it's just me, but I still don't understand how you estimate the lake depth. If the lake is empty and you measure from the bottom to the shoreline, I understand the method. But

what role does the ice dam play in this calculation? It is explained better in Chapter 2.3, but here the explanation is somewhat fuzzy.

Thank you for these observations, which have been explained well in the revised manuscript.

P8L198: Before the reference (Dillencourt et al. 1992) there is a "to" too much.
Corrected

P9L232: "regional exemplars" à "regional examples"
Corrected

P10L246: "resultant" à "resulting"
Corrected

P10L248: "for ease of trend comparison" could be better rephrased as "to facilitate the understanding of the trends"
Modified as suggested

P10L251: The authors mention four excluded outliers, however, in Fig.4, there are only two outliers plotted. I would suggest adding the two remaining outliers.

Note that outlier detection has been modified to be more rigorous, and there are no outliers.

P13L291: In Fig.5, the x-axis label contradicts the caption. Are the values the volumes of tetrahedrons or pentahedrons? In any case, the axis label should indicate that the volume values have been divided by 10 to match the measurements.
Caption and figure modified

P13L297: "improving" à "improved"
Corrected

P16L357: The abbreviation GLOFs has already been explained in L45.
Corrected

P17L359: Am I correct in the assumption, that the values in 6c are divided by 10 to match the measurements? Because this time, it is not mentioned in the figure or the caption.

The figure and caption have been revised to ensure there is no error in the interpretation

P17L372: "Values of values n' < were associated…" à There appears to be a missing value (probably "0.60") after the "<". Also, the beginning of the sentence should be rephrased. In the same line, there is a space too much before the new sentence beginning with "Therefore, in..."
Corrected

P17L378-380: While the discussion in general does very well in summarizing the article, this part does not belong here. It introduces information better suited for the Methodology as it justifies the

reason behind this geometric approach. In the next paragraph, there are a few more sentences with the same problem.

We understand the point the referee is making here, but it is important to remind the reader of the relevant methodological assumptions within the discussion. This way, the reader is guided through the Discussion rather than having to go back to the Methodology for orientation. Nevertheless, we have edited the Discussion to ensure a consistent connection between the elements discussed.

---

## Author Comment (AC3)

**Second Referees comments**

Review: Predicting the Risk of Glacial Lake Outburst Floods in Karakorum

Based on remote sensing as well as field data the authors aim to support their hypothesis that GLOFs can be anticipated from remotely sensed surge velocities as well as knowledge of lake geometries and lake expansion. They find a link between glacier velocity and lake volumes, and subsequently a relation between lake volume (or depth) and the risk of disastrous drainage. They do this based on 3 well known case studies in the Karakoram and in the end expand this to the global scale for the Discussion. The manuscript is in general well written and language is clear. Literature has to be revised at times and is not always completely appropriate with respect to basic conclusions in the Introduction, that are at the basis of the arguments then later followed. More problematic is that the hypothesis isn't clearly laid out at the start and hence the presentation of Results becomes a bit confusing, which would require some restructuring. Additionally, there is also a general lack of scrutiny on error margins especially for field measurements, which has implications on interpretation of results as I outline below, as these uncertainties propagate into your results. As I describe further down as well (and as the authors themselves admit throughout the text), 'prediction' itself isn't possible with this approach (if it will ever be) and hence the title is in my view misleading, and I would strongly consider to revise that.

Thank you for noting that the manuscript is generally well-written. However, as you suggest, some restructuring improves the flow of the arguments, and we have taken up your suggestions. As for the title, we have changed 'Predicting' to 'Anticipating,' which is more appropriate.

Most importantly however, I think that there remains lack of clarity on why surge velocities need to be part of the predictive mechanism and even the initial hypothesis that links them to the thickness of the ice dam. This oversimplifies a complex problem and the physical reasoning escapes me. This then also leads to strong interpretation of results, that are not as clear cut as the text makes them seem. All this in mind I think the manuscript needs a careful major rehaul before it is suitable for The Cryosphere (while the importance of the topic itself and the general approach I do think merit consideration for this Journal). I explain this in more detail with a number of major concerns below, followed by quite a short list of minor issues encountered throughout the text.

Thank you for raising the issue of glacier surging as a mechanism for predicting the formation of glacial lakes and subsequent GLOFs. We have ensured that the revised text does not suggest that glacier surges can be used to predict the formation of lakes. We had already noted that the physics of the problem is complex, and a complete understanding of the processes that enable the modeling of the controls on GLOFs is not imminent. Rather, some simplifying assumptions related to empiricisms would be valuable in providing guidance with reference to anticipating GLOFs. We have revised the text to ensure this is stated clearly.

 Major comments including conceptual/methodological issues:

Title: I am concerned that 'predicting the risk' is a very strong term as this will unlikely ever work to a degree that people understand what 'predicting' means and we shouldn't suggest we can

predict it (you say that yourself in the Discussion). You could change it to what you use later to 'Enhance the predictive capabilities for Glacial lake outburst risk in the Karakoram'

We have changed 'Predicting' to 'Anticipating,' as the latter word better relates to the requirement to provide timely warnings.

L14: Not all glacier snouts produce lakes and there is no evidence that proglacial lakes lead to advancements or surges! Please amend.

Thank you for noting this error, which was due to translation into English. The text has been revised.

L15: Recent research is conflicted on this topic and maybe actually showing the opposite - that ice dammed lakes drain less frequently (Veh et al., 2023). I would therefore remain very cautious with such kind of assertions to set the scene for your research.

The argument at this point relates to the Karakorum. However, we have modified the text to provide a caveat that GLOFs may decrease as global ice decreases.

L32: While mass gain has indeed been observed this trend is already over since approximately a decade – see (Jackson et al., 2023) for a number of studies referring to this. It would be prudent to note here that by now we consider this anomalous process to be over.

Thank you for the suggestion to provide a fuller picture of glacier behaviour, including outside of the period of time we are considering. We have amended the text, adding the Jackson reference, amongst others.

L34: The statement that there is a link between the anomalous mass balance and surges is spurious – what do you base this on (or which study specifically)? Also there has to my knowledge been no evidence for an increase in surge frequency (but only our ability to detect them).

It is indeed challenging to establish a definitive correlation between glacier mass balance and glacier surging due to the complexity of glacial dynamics and the multifaceted factors influencing both phenomena. However, it is overly strong to label the relationship as 'spurious.' Several studies provide evidence suggesting a link between mass balance and surge events. Bazai et al. (2021; 2022) have demonstrated that surges often follow periods of increased mass balance. Specifically, their research highlights that a higher mass balance can result in increased glacier velocity and surging activity. For example, Bazai et al. (2021) showed that the Shishper and Chilinji glaciers exhibited significant mass transfer from the accumulation zone to the ablation zone, leading to surges between 1999 and 2002.

Heïdi Sevestre (2015) and Bhambri (2017) also support this notion, indicating that surges are often associated with changes in mass balance. Their studies suggest that an increase in mass balance contributes to glacier thickening and enhanced basal sliding, which can trigger surging behaviour. Sevestre (2015) particularly emphasizes the role of mass redistribution within the glacier in surge dynamics. Additionally, Bhambri et al. (2019) observed that recent surges of the Shishper and

Chilinji glaciers involved the transfer of a higher mass compared to previous surges, indicating a possible correlation between increased mass balance and surge activity. While there is no conclusive evidence for an increase in surge frequency, improvements in detection capabilities might account for the perceived increase in surge occurrences. Enhanced satellite imagery and remote sensing techniques have made it easier to monitor and identify surge events that might have gone unnoticed in the past. This latter point is now noted in the Discussion.

L40: All three studies cited here for 'glacier avalanche increase' are based on few specific events, not trends (since so far we do not have long records). Hence you cannot speak of an increase of these events based on this evidence or need to adapt language to caution our lack of records so far.

This relevant text has been moved and now refers to an apparent increase. The issue of the lack of long-term records is better referred to within the Discussion, wherein we have acknowledged the lack of long-term records.

L41: I am not sure what you refer to when you speak of 'positive variation in regional climate feedback' with respect to lakes. You mean 'more lakes leading to more mass loss leading to yet more lakes'? Then spell this out, and you'd need to cite studies that show an increase in lakes leading to more mass loss (e.g. (Zhang et al., 2023)

Thank you for noting this error in English expression. The text has been completely rewritten.

L54ff: While (Carrivick & Tweed, 2016) is of course a global compilation, there is since a more recent overview globally (Veh et al., 2022) as well as for HMA specifically (Shrestha et al., 2023), both showing a much larger number of events and providing more accurate statistics for the numbers you are quoting here. Since we need to be mindful of the advancement of research, it would be prudent to refer to these updated studies here.

Thank you for the additional references that have been added to the revised text.

L96: 'Data' is generally not a subsection of Methodology but at least on equal footing. I would hence rename this section 2 to 'Data and Methods'. Also, you lack an introduction of your specific field locations and why you chose those but rather jump directly on the three lakes in L98ff. It doesn't require much, but an introduction of your field sites here briefly is required, possibly with an overview map as Figure 1 or an inset elsewhere. L146f actually is a kind of introduction in this regard that should be placed earlier.

We have renamed section 2 (now section 3) as 'Data and Methods' and sub-section 3.1 as 'Remote Sensing data'. A new section 2 has been added to the revised text to introduce the study area.

L167: Please explain how you 'estimated' this length.

We have added text to the revised method to indicate that lengths were measured using GIS 3D interpolation to UAV data and ground GPS points. These GPS points were then overlaid with the GIS 3D interpolation lines for enhanced accuracy.

L199: What are 'other' dimensions you get to with this method, please explain.

A reference to Fig 3b has been added to the text wherein the other dimensions are indicated.

L193ff and Figure 2: I am a bit confused by this section leading into results. You introduce the tetrahedral and pentrahedral shape of the dam, but never refer to the second but just the first. Why? You then also present results already in Figure 2b, even though we are still in Methods (especially your very last part on the issue of overestimated h). It is important to keep this apart and not mix the two sections.

Thank you for highlighting the confusion in the text at this point. We have restructured the text to ensure that the pentahedral approximation is detailed in the Method rather than just being introduced within the Results section.

 Finally, I think it would be important to rather use the space in methods how you were able to determine the D, E and Z (and Y, which is wrongly denoted twice as Y1 in the figure but should be Y1 and Y2!) in the field. How accurate are your GPS measurements? How accurately are you able to determine the position of the dam crest and base? The errors that are quite normal from such field measurements will then propagate into your volume estimates, which in a complex equation like equation 1 you show here, can become complex themselves. It is important to note them however to judge your comparisons you then make against volume estimates from the DEMs.

The two lengths Y are variable, but in Fig. 3b, they were drawn as both equal in lengths such that both lengths are denoted as Y1.  However, as Y can vary in both cases, we have changed the notation in Fig. 3b and made text changes to make the point clear. We have added text to indicate the maximum errors that might be associated with measurements obtained from remote sensing images or the estimation of lake depth from DEMs. For lakes typically 1 km in length, the errors are <3% and <14%, respectively. NB: The uncertainty in estimating the lake volumes using remote sensing, DEM, and UAV is already given in Table 1.

In turn, you also need to specify the uncertainty of your UAV derived DEM, which in turn translates into an error range for your volume estimate.

The error in the DEMs used was already reported in the original submission. The errors in lake volume determination were addressed in the prior reply to the reviewer.

L256f: You here come up with the physical hypothesis you try to prove after you present the results. That is confusing and should be the other way around, the hypothesis should be introduced in the Introduction already, when you set out to argue why you look at the datasets you choose.

We have broken the reviewer's comments down into sections to make it easier for the reader to follow. It was not our intention to introduce a hypothesis at this point rather than the text the reviewer refers to is an interpretation of the data within Fig. 4. Nonetheless, it would be better if hypotheses underpinning this interpretation were stated upfront, so we have introduced two hypotheses in the Introduction section, as follows:

"Consequently, herein, we explore the main hypothesis that lake volume is related to glacier velocity.  As lake volume can dictate the characteristics of a GLOF, a secondary hypothesis was addressed that ice-dammed lakes can exhibit geometries similar to regular geometric shapes, such that in the absence of a detailed lake volume date, lake volumes might be estimated from geometric consideration. "

I then however do not quite follow your hypothesis. You argue that ice thickness at the lake dam decreases as velocities of the surge peak. But lake dams are in the receiving zone of the surging glacier (mostly, definitely in all the three cases here), where velocity peaks coincide with thickening of the tongue. Ice thins in the quiescent phase after the surge (where again we see many GLOFs of course in all three cases, for many years to come). This leads me then to the main concern I have with the prediction plan based on just velocity here –

As the review infers, sites for ice-dammed lakes are formed during glacier surges when ice thickness increases. However, at that time, lake volumes were generally low as there was insufficient time for meltwater to build up, and crevasses allowed for ready drainage.  As the surge reduces, ice velocities reduce, and lake volume increases due to meltwater increments and the closure of crevasses. We are not suggesting that lake volume or GLOFs can be predicted using only one variable – glacier surge speed – only that there is a potential correlation that provides additional insight into lake formation and draining.

I would argue that lakes fill following closure of the subglacial drainage, from crevassing (as you argue) but also with changing lake water temperatures (depending on inflow temperatures and atmospheric temperatures and snow melt and glacier melt forcing. These variables are of course very hard to constrain, but with your approach you are packing these complex processes into glacier velocity alone – which is sometimes changing without any perceptible effect on lake properties as well. This way you gloss over other potential drivers, which could become a problem when wanting to be truly predictive. I think this is finally all reflected in your results.

We agree that changing water temperature may be a factor in lake growth, and the controls for these factors are complex. We make no attempt to suggest that these complexities are 'packed' into just one variable; instead, the speed in the velocity versus lake volume data actually reflects the role of these unquantified variables.  We have not 'glossed' over these other controls; many are noted in the Introduction. Rather, we are trying to present the evidence that one readily measurable control – ice velocity – may have an inverse relationship with lake volume. The importance of other controls was already acknowledged in the first paragraph of the Discussion.

Naturally there are GLOFs always after surges (Figure 3), but that just follows from the damming.

GLOFs do not always occur after surges, and surely, it is not satisfactory to state that they occur 'just' due to ice damming. Such a conclusion negates any attempt to understand the controlling processes better.

 I can't see any clear relation between speed up and occurrence just from these results. In Figure 4 you finally of course have a very large spread. For Shisper alone there seems to be no change of volume with velocity (I suspect here it is simply constrained by topography also since around later

drainage events the surge had stopped completely), at Kyagar the volumes varies widely with very similar velocities and I see no geometric relationship at all. It finally is a deficiency here that you do not show the uncertainty bars around velocities (especially since these mean values spread quite a lot during the surge itself on top of the error introduced from the SfM approach) or the volumes, which would further put your relations into context (and put the geometric relationship in question).

In presenting these results, we have taken a cautious approach, stating in the original text that the results are 'clearly not definitive'. In the Discussion, we acknowledged that the data scatter means that the trend line is not statistically significant. Nonetheless, the trend line is always negative, and we have suggested that further research might seek to better populate diagrams similar to Fig. 5 (formerly Fig. 4) to examine this relationship further.

L265ff: Maybe this is just me, but I would find it more easy to follow if you first present results on your ability to represent volumes of lakes with trigonometric considerations and then follow this by the much more complex relation with surge velocities. As it stands now you jump between one and the other, which leaves it quite hard for the reader to follow the final reasoning. When it comes to volume estimates you furthermore now have results spread between Figure 2 and Figure 5, with a completely different story in between. It would be advisable to bring this together to then come to a clear conclusion simply on your ability to estimate volumes accurately. Once that is done, it will also be more reliable to interpret your results from Figure 4. Relating to your volume estimates I am finally surprised that you do not refer to earlier considerations in this direction, e.g. taken in (Cook & Quincey, 2015), who summarize multiple attempts in literature in this direction.

Thank you for these observations. We have restructured the Results section to include sub-headings separating the surge velocity considerations from the trigonometric considerations. We have added the reference to Cook & Quincey, 2015.

L316ff and Figure 6: You have used multiple data to assess volumes/lake dimensions but do not specify what your lake geometries are based here now. They look very detailed, is this all taken from UAV generated DEMs and field measurements (for depth)?

The methodology was defined in the 'Data and Method' section, to which we have now added the uncertainty in both length measurements and the derivation of lake volumes.

It needs to be absolutely clear what data finally leads you to what conclusion, especially when you draw up ranges that may be used later for breach assessments but whose accuracy change widely depending on what data you base them on. I am also not entirely sure, how to interpret the red lines in Figure 6 – are these the actual depths/volumes at the respective GLOF events as presented in Table 1? Make that clear in the caption to Figure 6.

The red lines connect the elevation of the lake level at the closest time to the GLOF with the lake's volume at this time. To make this clear, text has been added to Fig. 7 (previously Fig. 6).

Also you then show relations for times when ice thickness wasn't well known right (e.g. older Kyagar events)? Doesn't you assumption of a constantly same geometry become weaker away

from your UAV-DEM date? You have deposition, sediment drainage as well as different dam heights every time. This will affect the final calculation of n/pressure and hence I suspect give you much more spread in 6d, which to me now seems surprisingly well lined up for Khurdopin and Kyagar.

Lake volume data are in Table 1 and, as shown in Fig. 7 (previously Fig. 6), are obtained from survey data and not from geometric approximation.

L328: Following from above to your conclusion that 500kPa is the threshold – this is simply 50 m of water table equivalent. Isn't that simply confirming that 'a lot of water is needed to cause a splash'? Below 50 m in most cases we will have small GLOFs (which have occurred often unrecorded in all lakes and one could argue aren't really GLOFs but simply high flow drainage events) while when you get to 50 m you slowly get to volumes that cause considerable flow. But why would that be specific to surging glaciers or the region? Again this will then depend on ice dam properties (temperature, thickness) which bring in a number of other unknowns and can't just be summarized with a single value. Even your results show that for Shsiper it may be >1000 kPA, while for Khurdopin 700 kPA, quite big differences.

The pressure on the ice dam face derives simply from the water depth at the dam face. We have not sought to determine that 'a lot of water is needed to cause a splash' as the reviewer suggests. Instead, in the initial submission, as in the revision, we have sought to provide some indication of the relative depth (new Fig. 7d) at which GLOFs occur, such that warnings might be provided downstream in a timely fashion. There is nothing in the literature to indicate what might be considered a minimum sudden discharge to be labelled a GLOF. We have noted that a few of the recorded small discharges occurred well below a potential threshold of non-dimensional depth above which substantial GLOFs have occurred in the Karakorum (new Fig 7d). In the text, we have made no case that such behaviour is specific to the surging glaciers in the region, so we are unsure why the referee raises this point. Rather, as the referee acknowledges below, we have extended the analysis to include other glaciers worldwide.

L340f: It is nice that at this stage you manage to pull in examples from elsewhere to support your volume/threshold theory. This of course gives your approach some strength. This leads however to two observations:

- Maybe it would be prudent to leave away the surge aspect from your study altogether, to me the velocity story doesn't quite hold up and also does not have a clear connection to the critical threshold. It even further weakens your prediction capabilities as making the step from 'surge velocity u' to 'critical threshold n' becomes spurious and doesn't hold as a clear indication (i.e. you can't say 'if Shisper exceeds a velocity of x m/d a GLOF may be more likely').

The reviewer makes some useful points here. However, we have not tried to relate the surge speed directly to the critical depth, so we do not understand why the reviewer would suggest that we have. We argue cautiously that there might be a relationship between surge speed and lake volume and indicate that further work is required to test this idea.

- As you note yourself in L391f, you can't 'predict' above a certain threshold (hence also my concern with the title), you can just indicate higher likelihood of catastrophic drainage. This itself is useful but the recommendation to always report that as imminent threat has its drawbacks, as people will become less sensitive to warnings if they are reported often without any resulting events. Hence I rather suggest to use this as a general support in risk assessment procedures (paired with other means).

We accept that 'predict' was too strong a word and have changed it to 'anticipate' in the title and in the main text. We acknowledge that human nature can mean that warnings are not considered by the general public, but the reviewer seems to be implying that there is no point in issuing warnings. Whether the imminent risk is considered within risk assessment procedures or used to directly warn people would be based on specific countries' own procedures.

- In L380 you get back to the volume estimation – I agree that for a first order measure this is useful, but to make this a considerable improvement over simply seeing lake area from space (much easier), the reader requires much more scrutiny on your volume estimates. How accurate are the depth/dam dimension variables you have to come to good volume estimates here and how does that propagate to your n computation to make it robust? This is currently missing from the text and with the data you show from lakes elsewhere could be expanded upon for a thorough Discussion.

The resolution of the remote sensing and the accuracy of depth measurements were quantified in the original submission. In the revision, we have added text to clarify the uncertainty in lake volume estimates, which is already stated in Table 1. It is well known that linear measurement errors are additive or can cancel out. However, we have added text to the Discussion to emphasize the fact that mensuration errors can now be quite small given modern techniques.

L410f: The Conclusion as it stands is too short. It also hinges much weight on the linear regression against one means to estimate volume, while I do not think that was the initial aim of the paper (prediction and GLOFs rather than lake volumes). This suggests that the general aim needs sharpening and the conclusion should also converge on that. From your Conclusions I would also like to see to what degree your field data was essential in this process as this clearly impacts the potential to upscale. For this a more thorough descriptions of said field data including their uncertainties is however required (even if parts of this has been presented in other publications).

Thank you for these helpful comments. We have restructured the Conclusions, putting the geometric estimation conclusions after the comments on the lake draw-down. We have added a comment on the value of field data. The uncertainties of the field data are detailed in the Method.

Minor:

L1: While 'Karakorum' spelling is of course possible, I would consider going with the standard in English in literature by now and go for 'Karakoram'

We have standardized spelling to 'Karakoram'.

L40: 'has increased'

Text altered.

L92: '…research foci described in this study aims to leverage …'

Text altered.

L142: 'measured'

Corrected.

L147: 'have occurred'

Text altered.

L150: 'Sentinel 2', capitalized

Corrected.

L165: 'the empty lake'

Text corrected.

L197: 'outlines'

Corrected.

L198: remove first 'to'

Corrected.

L205: 'a value which can be …'

Text altered.

L240/Figure 3: typo in the figure – 'Kayager'. Also what are the numbers in 3d? I assume days after 1/1/1990? You would need to say that but it is also a very hard way of reading your data. Dates would be much more useful.

Kyager has been corrected, and the figure panel has been redrawn. However, adding dates would cramp the lettering on the x-axis.

References:

Thank you for the additional references which we have cited in the revised manuscript.

---

## Author Comment (AC4)

**Third Referee Comments**

Thank you for the opportunity to review the manuscript "Predicting the Risk of Glacial Lake Outburst Floods in Karakorum" by Bazai and co-authors.

In their study, Bazai et al. ask the important question of when an ice-dammed lake reaches a critical depth, requiring further investigation and possibly warning of impending flooding. This is a timely question, as many ice-dammed lakes form and drain each year in this and other regions. However, the text lacks structural clarity and motivation as to why we need both geometric models and DEMs to better understand the volumes and depths of ice-dammed lakes.

Thank you for noting that the subject matter is timely. The text structure has been comprehensively overhauled to address the referee's concerns regarding clarity and motivation.

Parts of the results should be better placed in the methods to motivate the two geometric approximations developed to estimate volumes and depths of ice-dammed lakes.

Agreed, as noted above, restructuring has addressed this issue.

The referee noted that the text at the original line 69 "needs to be rephrased to something like 'can lead to...' or equivalent. In most of the regions mentioned, most contemporary ice-dammed lakes form because a tributary becomes ice-free, and the main glacier trunk then impounds part of the meltwater from the tributary catchment. How many glaciers in Norway, the European Alps or Patagonia have been reported to surge and form a lake in past decades?"

The sentence has been rephrased to indicate that glacier surges can result in ice-dammed lakes in these regions. As the focus of the manuscript is on surging glaciers, there is no need to note other means by which ice-dammed lakes can form. We have acknowledged that there are other means of forming lakes.

At original line 76 the reviewer states " please be more specific what knowledge gaps remain. Otherwise, this statement would downplay decades of research on ice-dammed lake formation and drainage in Iceland, Norway or Switzerland."

We noted the range of topics for which information is lacking earlier within this paragraph. It is not our intention to review the details of where knowledge is deficient within this submission. Rather, we had indicated the focus of the study within our hypotheses. We have not downplayed the significant contribution of previously published research within this area.

 The idea that surge velocity controls lake depth is interesting, but the underlying analysis falls a bit short, using mostly visually guided drawing of exponential curves.

We agree that the relationship between surge velocity and lake depth is not clear-cut. However, these data are hard-won and worth reporting, as we wish to stimulate further focus on the potential for a conclusive relation to be shown in due course. We have used suitably cautious language in reporting and discussing these observations.

Another good point is the comparison of ice-dammed lakes and their depths at failure in other regions, but there is no information on how this data was obtained and processed.

The details of how these data were obtained were included in the original Method section. However, we have edited the Method section to ensure this information cannot be overlooked by readers, as it was by the reviewer.

The discussion is largely focused on a single paper (Carrivick et al., 2020), but could benefit from a more thorough reflection on previous work on the geometry and processes involved in the drainage of ice-dammed lakes.

Thank you for your feedback. We appreciate the opportunity to improve our manuscript. In response to the reviewer's suggestion, we have expanded our discussion to incorporate a broader reflection on previous work concerning the geometry and processes involved in the drainage of ice-dammed lakes. Below is an overview of the additional references we have included and their contributions to the topic: Zhang et al., 2023, Walder, J. S., & Costa, J. E. (1996), Ng, F. S. L., Liu, S., Mavlyudov, B., & Wang, Y. (2007), (Zhao et al., 2017), Richardson, S. D., & Reynolds, J. M. (2000), and Quincey and Luckman, 2014.

Please see the attached PDF for more specific comments.

The marked manuscript comments by the referee were copy-protected, so we cannot copy them verbatim on this document. We have addressed all the comments made by the referee on the marked manuscript s/he provided, adding references as required. Here, we reply to the more substantive comments made by the referee flagged by the original line references in the manuscript. If there is no reply to a specific comment, it is because it was minor and we have made a suitable alteration.

Hazard v. Risk. We have modified the title as suggested and checked the manuscript for the correct use of these two terms.

Abstract: Minor comments made by the referee have been dealt with within the min text as there is no space to amplify the points within an abstract.

Introduction: We have emphasized the focus on ice-dammed lakes rather than moraine-dammed lakes. We have rewritten the text to make it clear that most glaciers in the HMA are now in recession, and we have added the references the referee suggested.

The reviewer indicates that the purpose and goal need to be outlined better. The objective of the study is defined in the final sentence of the Introduction.

The reviewer suggests that the study sites need to be introduced better, perhaps with a small chapter on its own. We have added a separate section after the Introduction and before the Method sections.

please phrase clearly. Reads as if 23 different lakes formed, but I think, it's three lakes that formed and drained repeatedly (23 times) in a given period.

We understand the reviewer's point. However, we cannot argue that only three lakes drained repeatedly. The glaciers are surging, and lakes form and drain repeatedly. Whether they

represent the same local conditions, such that they could be regarded as the 'same' lake each time, is a moot point.

The reviewer asks the questions: "so, the inlet is where the bed elevation is lowest and where the glacier is thinnest? could that be a consequence of a collapsed tunnel?". We cannot speculate on this matter, and, indeed, the text referred to is within the Method, so it would not be appropriate to make further statements in that section.

In our statement regarding the distortion of UAV images, the referee asks, "What does elevation have to do with image distortion?" There are many image issues related to the altitude at which UAV imagery is obtained. The best-known is parallax, but other than noting that we minimized distortion, there is no need to go into the technical details here.

The reviewer notes that there is "no info about the (digital) processing chain to generate DEMs and orthophotos?" We supplied references herein for succinctness. The DEM procedure is detailed.
Bazai et al. (2021) and Gardelle et al. (2013).

The reviewer states "if you have a DEM that shows the lake empty - why do you need a geometric approximation of the lake depth and volume then? couldn't you just fill the DEM until you reach the elevation of the manually mapped lake outline?"

Thank you for noting this issue. We have revised the text to clarify that the procedure is developed for situations wherein the lake is not drained.

The reviewer asks for more information on the characteristic lengths of lake basins and the procedure for determining lake volumes. This section has been fully revised to take these points into account.

The reviewer states "I read the assumptions about the lake geometry several times. It is important emphasize and justify these assumptions better, maybe using a table that show relations between lake width and depth from measurements that you extracted from the DEM."

Our assumption of lake shape was based on visual assessment of UAV and satellite images; we have revised the text to make it clear that, to ensure an objective approach, we estimate the lake volumes assuming geometric considerations and only then compare the volume with those measured from DEMs.

The reviewer notes that the message within the following sentence is unclear. "In passing, it can be noted that the volumes of the irregular tetrahedrons in most cases were not dissimilar to the volumes of regular tetrahedrons (*i.e.*, triangular-based pyramids), the equation for which is simple, in contrast to Equation 1."

We agree and have deleted this sentence.

In the method statement regarding assuming geometric shapes for lakes, the reviewer asks "why we do not introducing a constant that scales h to the 'true' depth of the lakes?"

We are applying the method to lakes that have not drained and for which there is no depth information. The revisions to this section have clarified the method.

At the end of the Method section the reviewer states: "n the abstract, you write about n' and the pressure in the lake necessary to generate an outburst flood. why is there no emphasis in the methods how to obtain these quantities?"

The significance of the value 'n' is only realized within the Results and is considered fully within the Discussion. It would be confusing to introduce n in the Method as it is not part of the method but an outcome of the results. Water pressure is a direct function of water depth, so it would be trivial to explain how depth and pressure are related.

In line 248, the reviewer asks, "Does the repeat rate of Landsat images allow for obtaining daily glacier flow velocity? I think this is probably again an average?" Yes, the velocity obtained from Landsat images represents an average velocity. This is calculated by measuring the glacier's displacement between two available images and dividing it by the number of days between those images. Consequently, the derived velocity is an average over the period between the two images.

At line 256, the reviewer asks if we extrapolated the valley walls downwards to determine whether the valley is V-shaped or U-shaped. We state that we did, and so that is a fact. We did not provide any analysis of this procedure, as the decision of V-shaped versus U-shaped is inevitably subjective. Rather, the statement here is to remind the reader of the concept of viewing the geometry of a lake as an irregular tetrahedron or isosceles triangle.

With reference to the shape chosen to represent the lakes the reviewer asks "Why not semicircular? And the lake being represented as a half cone?".

It is possible that other geometric shapes might be chosen to represent the shape of lakes. We did not explore further shapes as the chosen shapes give a good comparison with the volume of lakes obtained using DEMs, as is shown within the Results.

I understand that the exponential relationship suggests both, that depth reduces as surge speed increases, or that depth increases as surge speed reduces. However, I am not sure whether your statement is generally valid, because throught most of your time series, you have zero depth (because there is no lake), but low glacier flow velocities.

The point made is valid if we were considering the full period of investigation. However, the velocities used to produce Fig. are the average velocities for each surge period, so these values do relate to the period of lake formation.

With reference to the exponential functions the referee notes "this sounds tempting, but given the more visual fits of the exponential function, I would refrain from this statement."

Thank you for this cautionary note. We have revised the analysis and interpretation of Fig. 5.

At former line 298, the referee asks how much depth must be reduced to ensure that the lake volume is defined by an irregular trihedron. This value will vary lake by lake, so it is not useful to define any such value. Rather, we have indicated that regression functions between

the assumed geometric lake volume and the measured lake volumes (from DEM) are the appropriate comparators.

The referee states "I still have a hard time to understand why you fit the geometry model, if you can infer the lake volume directly from the DEM.

We have reworded the text to clarify that the geometric procedure was developed to determine whether the lake's volume can be approximated accurately. The volume from the DEM is required to complete the test.

The referee asks why the relationship between the "roughly half" tetrahedral volume and the DEM volume is not illustrated. The reason is that this information is not required graphically, as the "roughly half" relationship indicates that a quadrilateral approximation might be better. This explanation was provided in the sentence following the referee's query.

again, this equation and reasons for assuming either lake geometry should be part of the methods. It's a lot going back and forth between theoretical considerations and the results.

Agreed. This equation and the explanation for its use are now found in the Method section.

The referee asks, "How did you arrive at estimating the (hydrostatic) pressure?". According to physics, the water pressure is directly related to the water depth.

The referee notes that the "discussion is not really rich of literature. The idea that lakes drain at a certain critical depth has been developed by Thorarinsson in 1939. What is the novelty compared to his findings?"

We were aware of the Thorarinsson paper but did not cite it for reasons explained here. However, we have now acknowledged his contribution in introducing a concept of critical depth by citing the relevant paper. Thorarinsson introduced the concept of a "critical zone" (k) which he designated (in his example) as having only one elevation (500 m), which is not a zone. He refers the reader to his Fig. 3 for the definition of k but k is missing in Fig. 3. He also assumed that a correction, for crevassed ice, can be applied to lower the ice barrier height and finally he calculated the height of the ice barrier relative to the lake level that would allow the ice barrier to lift (*i.e*, float). An ice barrier cannot 'lift' unless the pressure is equalized on both sides (*i.e.*, there has to be water on both sides to provide lift for the whole ice barrier. Given the uncertainty of Thorarinsson's assumptions, we can see no value in discussing our definition of *n'* with Thorarinsson's k as we are unable to come to any meaningful outcome.

CONCLUSIONS. The referee asks for a reference to be added to the Conclusions to support the sentence "Year after year, there is a rise in human casualties and losses to residences, infrastructure, the energy sector, and local and international trade.". It is not usual to include references within the Conclusions and suitable references were included within the Introductory section.

FIGURES CONSIDERED BELOW HAVE THE NUMBERS UPDATES

Fig. 1 The reviewer suggests adding the ownership of the images herein. However, the authors are the owners so there is no need to provide a credit.

Fig. 2. The reviewer suggests that we "could be good to use capital letter for coordinates, and lower-case letters for edges and lines."

The problem with this suggestion is that we only have two coordinates (A, B), and it is logical to use this notation. Using lowercase 'a' in the main text could lead to confusion.

Fig. 3 With reference to the new Fig. 3a figure, the reviewer asks, "how does Fig 5 show that the edges of the triangular lake have the same length?"

We are unsure why the reviewer asks this question as the figure does not show a lake with equal side lengths.
Fig. 4a the difference between the red and blue velocities is explained in the revised figure caption.

Fig. 4d. The reviewer states it "would be good to emphasize the timing of the GLOFs more. I am not sure whether I see a very systematic pattern in Figure 03a-c." S/he also queries what the bars show and what the horizontal axis represents.

We are unsure how to address the point of emphasis (within the main text) without laboriously comparing surge timing with each individual GLOF occurrence. The basic data are in Table 1, and Fig. 4d shows the timings in a relative sense, which should suffice to make the point. The bars were defined in the caption, and the horizontal axis has been revised to make its relationship to time evident.

In the main text, we suggest that lake volumes would increase when the ice speed is low. The referee questions this statement: "In a number of cases in Fig 4, the GLOF seems to occur around the peak glacier flow velocity."

The referee suggests that the hypothesis that "surge speed should control lake depth, volume, and potential GLOF volumes, and states "isn't this hypothesis very much dependent on the exact position of the lake? At the end of the surge phase, redistribution of the ice mass may cause a lower ice thickness in the middle part of the glacier, but a thickening at the glacier terminus. This is what I see when looking at Fig. 10 in Round et al. (2017)."

We acknowledge that there may be confounding factors influencing this hypothesis, yet the purpose of Fig. 5 is to explore the statement. We anticipated scatter in the data due to factors such as the referee indicates and, in the Introduction, we had already noted the complexity of the multiple factors which might pay a role. Thus, our proposition is a general one based on Fig. 4d. We have already acknowledged that there can be exceptions, and the issue of the controls on ice-dammed lakes is complex. At this point, we are introducing the possibility that surge speed might be one of the multiple controls on lake formation.

Fig. 5. The referee queries the exclusion of outliers in the regression analysis and the fitting procedure for the curves. The outliers were excluded using statistical procedures. However, we have revised the text and included all data points in the regression analysis. The fitting of curves is now explained more clearly. There is no upper limit to the exponential function.

With reference to Fig. 5, the reviewer asks, "does it really make sense to throw lakes of different (ABSOLUTE) sizes into this model?

Fig. 5 shows the data for all three glaciers visually. The data are not lumped into one analysis. The regression function is fit only to the Kyager data as this is the only 'large 'data set.

Considering a small lake, possibly because the thickness of the ice dam does not allow the lake to grow more: what does the surge speed might tell you then compared to really thick surging glacier?
Wouldn't it make sense to talk about RELATIVE lake sizes in terms of the lake volume relative to the maximum storage capacity of the lake?"

These are useful questions, but there is currently not enough data to examine them with any confidence. Developing an analysis based on relative lake sizes is further problematic as the maximum storage capacity is unknown.

Again, with reference to Fig. 5, the reviewer asks, "Could it make sense to plot the volume against the duration since peak glacier flow velocity? I guess it takes some time for a lake to form as soon as ice motion decreases.

We agree there are alternative approaches to relate glacier behaviour to lake volume (if indeed there is one). We believe that there may be a relationship such as the one shown in Fig. 5 and have presented the data in this format to encourage others to explore the possibility of such relationships, which might, as the reviewer suggests, include lag effects.

Fig. 6. The referee asks what DEM was used to produce the long profiles.

The DEMs for Shishper and Khurdopin are derived from UAV-generated DEMs, while the corrected SRTM DEM and the ALOS PALSAR DEM were used for Kyager Glacier.

Fig. 6. The referee asks which panel is for the Kyager glacier.

The caption indicates that Kyager is panel c.

Fig. 7. The referee notes, "this is a really interesting figure; however, you did not introduce the reader to the underlying data and how you obtained them."

This is a summary figure and is based on i) the data obtained during this investigation or ii) extracted from publications. The method used to obtain data for (i) is provided within the Method section, and the Results are summarized in Table 1. Data for (ii) are extracted directly from publications.

Fig. 7. The referee asks "why is the elevation above sea level important? isn't the elevation relative to the lake level more important?"

This is a good question. The absolute elevation is used as a convenience as it neatly separates the bar graphs to avoid overlap. Relative elevation is more appropriate for analysis and is subsumed in the parameter $n$'.

Fig. 7. The referee suggests that we "Use an informative title for the figure, please."

The figure caption has been rewritten.

The editor has asked for a third opinion on this manuscript. Admittedly, I can only add a few points to the comments of the other two reviewers, who have done a very good job in suggesting ways to improve this manuscript. If their comments are fully taken into account, I expect that this manuscript can be a suitable one for the community focusing on ice-dammed lake outburst floods.

Thank you for noting that if the revisions are thorough the work should be publishable.

---

## Referee Report (RR1)

Review of the revision of the manuscript:

**Anticipation the Hazard of Glacial Lake Outburst Floods in Karakoram** by N.A. Bazai et al.

The authors of the manuscript "Anticipation the Hazard of Glacial Lake Outburst Floods in Karakoram" took great care to respond to the reviewers' comments. The article now reads better than before, the reading flow of the individual chapters is much more structured and many ambiguities have been resolved. However, one main problem remains: The focus of the paper and the objective are still not entirely clear. The title is too vaguely formulated and can be interpreted as pointing in the wrong direction. A "GLOF hazard anticipation" for the Karakoram is not the topic of the paper, it is much more focused on the relationship between surge speed / critical lake depth and GLOF events.

I have summarized the comments of the three reviewers and discuss how I think the authors have implemented the suggestions. If the remaining minor revisions are incorporated and the title and aim of the paper are coherent, I recommend publication, as the topics covered are quite interesting and of value, especially if the parameters for water pressure and critical depth are incorporated into a GLOF warning system.

General comments:

1) *The title and the aim of the manuscript*
The authors have replaced "predicting the risk" with "anticipating the hazard", but a true hazard analysis would need to consider more parameters than just lake volume and surge velocity (e.g., the stability of the ice or moraine dam, the likelihood of triggering events, or the potential paths of the flood). Therefore, the title is too broad and does not reflect the content of the manuscript. Also, mentioning the Karakoram region in the title suggests that the results are somehow specific to this region (apart from the location of the lakes, which is not relevant in the manuscript) – which, as the authors admit in their response, is not the case. I would therefore suggest that the title be changed to better reflect the scope and aim of the manuscript. A title along the lines of "Refining Lake Volume Estimation and Critical Depth Identification for Enhanced GLOF Event Anticipation" would better reflect the authors' statements at the end of the introduction and in the conclusion.

2) *The topic of surge velocity*
Following the reviewers' comments on the relationship between surge velocity and GLOF event, the authors have made some changes. However, it is still not clear enough to the reader why this topic is part of this manuscript. The introduction states that the "primary objective of this study is to improve the predictive capability of GLOF event timing by refining empirical lake volume estimates and identifying critical depths for future hazard and risk reduction". Surge velocity is not mentioned – which is not a problem in itself, as other papers have investigated this at the same lakes (e.g., Bazai et al. 2022, Li et al. 2023). The authors themselves warn that this part of their results should be treated tentatively and with caution. In my opinion, the more valuable results are the surprisingly accurate geometric estimates of lake volumes and the investigation of critical lake depths. I would therefore agree with the other reviewers in suggesting that the part on surge velocity be omitted.

Specific comments:

1) Chapter 2, Study Area, is a welcome addition to the manuscript and serves its purpose well. However, it lacks a review of previous research for the selected glaciers and their lakes. There has been some research focused on these three glaciers and their lakes, some even focusing on the relationship between surge velocity and GLOF occurrence. For example, Round et al. (2017) state that surges are the main factor in lake formation at Kyagar Glacier, while Li et al. (2023) found that the volume of the lake there reaches its maximum three years after a surge period. Furthermore, Bazai et al. (2023) conclude that surge velocities have a significant impact on lake formation at Khurdopin. I would therefore encourage the authors to add some paragraphs detailing previous research in order to avoid duplication in later parts of the manuscript.

2) Regarding Figure 1, I have some minor annotations, mainly for the overview panel in the top right: The fonts are too small to read – and not really necessary, as the other regions of HMA are not part of this study. Also, the lon-lat grid is too small, and also not necessary for this kind of map. Thirdly, I understand that the map depicts a hillshade of the region, but the color map is unusual and would need to be explained. I suggest using a surface reflectance with low opacity and an underlying hillshade would be easier to understand, but that my only be my opinion. In the three small panels, the authors indicate a "Dam site". Maybe I'm misunderstanding something, but isn't the location of the ice dam at the very other end of the lake where the lake connects to the glacier? The arrows seem to point to the inlet of the lake.
At last (and again, this is just personal preference): Please make the outline of the three glaciers stand out from the outlines of all other glaciers. If you choose to include the Indus, please use a shapefile with a higher resolution. And to be *very* nitpicky: Please adjust the scale bar lengths to 100 km (panel a) and 1000km (overview panel).

3) In Chapter 3.3 (L197f), the authors claim that the inlet of the subglacial conduit is always located at the deepest position in the ice dam. This may well be true for most cases, but the possibility of higher drainage channels in the ice should be mentioned, as this would have a large influence on the potential GLOF volume. I would also advise the authors to rephrase the start of this paragraph (L195) because, contrary to their statement, there are records of higher drainage channels at Kyagar Glacier (Li et al. 2023).

4) As mentioned in the general comments: Chapter 4.1 is very long, although the reviewers have criticized the presentation of the surge-GLOF relationship. Although the authors have partially addressed this, the chapter still adds little to the scientific discourse. The results presented are tentative at best, but there are other studies that clearly define the relationship between surge events and lake development as an increase in lake volume in the months/years after surges, i.e., a negative relationship between velocity and volume. The focus of this paper should be on estimating of lake volume and critical water pressure/lake depth, not on the relationship between surge velocity and lake formation. I agree with the other reviewers in suggesting that the paper be restructured and the surge aspect de-emphasized in order to properly focus on the more original content.

5) The conclusion is indeed very short, as previously mentioned by the reviewers. Furthermore, the first five lines are not really part of the conclusion but more of a secondary introduction to the topic. To my mind, the conclusion should comprise the summary of the findings (a little bit more detailed than in its current state) and give indications on how to use them (which are currently absent). The ability to apparently determine the lake volume quite accurately based on the geometric estimate is an important point and should be the focus of the paper, together with the critical depth parameter. In the conclusion, the authors should describe in more detail how and when these values can be applied meaningfully.

Technical corrections:

L 165: "Khurdophin" → "Khurdopin"

L 206: The layout of Table 1 seems very crowded. If the need arises, the column "Type of drainage" could be omitted, because it mirrors the values in "Vol after".

L 261: The assumptions are not addressed in the Discussion, but in the Results.

L 264: "ice dam volume" → "ice-dammed lake volume"

L 281: For the sake of completeness some comments on Figure 4, despite my concerns with the whole surge analysis in general and chapter 4.1 more specifically: My main point is that panel d does not correctly summarize the other three panels. The bars representing the surge periods do not correspond to the position of the GLOF events. One example: The first four GLOF events at Khurdopin appear during the surge period in panel a, but in panel d they all appear after it. The same applies for multiple other occasions. Also, the labeling of the X-Axes is still confusing, as the labels area sometimes five and sometimes six years apart – in the same panel. Why is the X-Axis in panel d formatted differently when the style of the other panels could be used here as well? Why doesn't the X-Axis in Panel c start at 0 as well?

L 301: This figure is a response to a reviewer's comment, but since it takes up a lot of space, I would suggest moving it to the supplementary material.

L 318: In this chapter and the next, the authors assume the lake surface to be a triangle. This is only the case for the lake at Shishper Glacier. I would encourage the authors to address the question how their method can be applied to more unregular surfaces, like the lakes at Khurdopin and Kyagar.

L 335: In the corresponding figure, this coefficient is given as 0.93. Which is correct?

L 347: "of the lakes was determined…" → "of the lakes determined…"

L 349: The authors need to provide references for their statement that the lake depth is the main driver for tunnel drainage. According to other studies, low water pressure inside or under the glacier can also trigger a drainage.

L 355: The authors should choose their words more carefully when describing the applicability of their n'-value. It could play a valuable role in determining GLOF probability, but while they do not say so explicitly, readers could gain the impression that with a lake level of n'=0.6 there is the danger of a GLOF occurring imminently. It should be made clearer that a GLOF warning would be based on more components than lake depth.

L 363: Please be more specific here: Low values of n' do not necessarily indicate shallow lakes, as it is a relative value. A large lake with a maximum depth of 300m could still contain a significant amount of water with a value of n'=0.5.

L 363: As one reviewer mentioned before: The reasoning behind the 500kPa is not clear and may confuse the reader. You have three glaciers with three significantly different water pressures – why choose the Kyagar? Do you mean to suggest that lakes shallower than 50m (~500kPa) are not capable of producing significant GLOFs?

L 364: I agree with the other reviewers: The regional aspect of the values for depth and water pressure is not clear. Why is the region mentioned here?

L 391: Again – why are your results specific to the Karakoram region? Even the three glaciers selected for this study are very different in terms of the shape, area, depth, and volume of their lakes – which of these characteristics are typical for this region?

L 419: The value of 0.6 appears solid in this context, but the second part of this sentence should be rephrased: the lake depth would be part of the hazard determination, and in itself not sufficient to issue a warning.

L 422-424: This should be mentioned before and not be part of the discussion, as it does not discuss results.

L 451: A small point: n'=0.6 does not automatically mean 510kPa, as n' depends on the water depth, while 510kPa corresponds to a fixed depth of approximately 50m. If n' is only valid from a water depth of 50m, this should be mentioned in the paper.

References:

Bazai, N. A. et al. (2022) Glacier surging controls glacier lake formation and outburst floods: The example of the Khurdopin Glacier, Karakoram, Glob. Planet. Change, 208. 10.1016/j.gloplacha.2021.103710.

Li, G. et al. (2023) Characterizing the surge behaviour and associated ice-dammed lake evolution of the Kyagar Glacier in the Karakoram, The Cryosphere 17(7), 10.5194/tc-17-2891-2023.

Round, V. et al. (2017) Surge dynamics and lake outbursts of Kyagar Glacier, Karakoram, The Cryosphere 11(2), 10.5194/tc-11-723-2017.

---

## Author Response (AR2)

**Editor Comments**

**Public justification (visible to the public if the article is accepted and published):**

Dear Nazir Ahmed Bazai and Co-authors,

The reviewer who reassessed the manuscript has now submitted his feedback. The reviewer notes that most of the comments have been addressed; however, there are still some issues that need to be revised. For example, the focus of the paper is still not clearly outlined, a point raised by all three reviewers in the first review. Furthermore, the title suggests a focus on GLOF hazards, but this is not fully reflected in the article. I kindly ask you to consider the comments and suggestions (see attachment). A final review of the revisions will then be done by me. If all issues are addressed, the article will be recommended for publication.

Yours sincerely,

Tobias

Thank you for your comments. Our replies follow the specific comments of the reviewer below.

**Referee Comments Addressed Point by Point**

Dear authors,

In your contributions, you have endeavored to incorporate the minor and major revisions of the three reviewers into your article, and you were largely successful in doing so. The article now reads better than before, the reading flow of the individual chapters is much more structured and many ambiguities have been resolved.

However, one main problem remains: The focus of the paper and the objective are still not entirely clear. The title is too vaguely formulated and can be interpreted as pointing in the wrong direction. A real "GLOF hazard anticipation" for the Karakoram is not the focus of the paper, it is much more specific.

I have summarized the comments of the three reviewers (see pdf) and discuss how I think the authors have implemented the suggestions. If the remaining minor revisions are incorporated and the title and aim of the paper are coherent, I recommend publication, as the topics covered are quite interesting and of value, especially if the parameters for water pressure and critical depth are incorporated into a GLOF warning system in the future.

The pdf is attached.

Text below is that found in the reviewer's pdf

Review of the revision of the manuscript:
**Anticipation the Hazard of Glacial Lake Outburst Floods in Karakoram** by N.A. Bazai et al.
The authors of the manuscript "Anticipation the Hazard of Glacial Lake Outburst Floods in Karakoram" took great care to respond to the reviewers' comments. The article now reads better than before, the reading flow of the individual chapters is much more structured and many ambiguities have been re-solved. However, one main problem remains: The focus of the paper and the objective are still not entirely clear. The title is too vaguely formulated and can be interpreted as pointing in the

wrong di-rection. A "GLOF hazard anticipation" for the Karakoram is not the topic of the paper, it is much more focused on the relationship between surge speed / critical lake depth and GLOF events.

I have summarized the comments of the three reviewers and discuss how I think the authors have implemented the suggestions. If the remaining minor revisions are incorporated and the title and aim of the paper are coherent, I recommend publication, as the topics covered are quite interesting and of value, especially if the parameters for water pressure and critical depth are incorporated into a GLOF warning system.

Thank you for recommending publication subject to minor changes. The referee notes that "The focus of the paper and the objective are still not entirely clear.". At the end of the Introduction (Section1) we provided our objectives as follows "Herein, the primary objective of this study is to enhance predictive capabilities regarding GLOF event timing by refining empirical lake volume estimation and identifying critical depths for future hazard and risk reduction.". However, we have now edited this statement to ensure there is no confusion and that the role of glacier surging in controlling lake volume is also acknowledged as an objective, as follows: "Herein, the primary objective of this study is to enhance predictive capabilities regarding GLOF event timing by refining empirical lake volume estimation and identifying critical depths for future hazard and risk reduction. We seek to achieve the objective within a framework of adjustment of lake volume to glacier surge speed, which has implications for changes in the depth of lakes relative to the heights of the ice barriers that impound the lakes."

General comments:
*1) The title and the aim of the manuscript*

The authors have replaced "predicting the risk" with "anticipating the hazard", but a true hazard analysis would need to consider more parameters than just lake volume and surge velocity (e.g., the stability of the ice or moraine dam, the likelihood of triggering events, or the potential paths of the flood). Therefore, the title is too broad and does not reflect the content of the manuscript. Also, mentioning the Karakoram region in the title suggests that the results are somehow specific to this region (apart from the location of the lakes, which is not relevant in the manuscript) – which, as the authors admit in their response, is not the case. I would therefore suggest that the title be changed to better reflect the scope and aim of the manuscript. A title along the lines of "Refining Lake Volume Estimation and Critical Depth Identification for Enhanced GLOF Event Anticipation" would better reflect the authors' statements at the end of the introduction and in the conclusion.

Thank you for these considerations. We have accepted the reviewers suggested title and have made the change.

*2) The topic of surge velocity*

Following the reviewers' comments on the relationship between surge velocity and GLOF event, the authors have made some changes. However, it is still not clear enough to the reader why this topic is part of this manuscript. The introduction states that the "primary objective of this study is to improve the predictive capability of GLOF event timing by refining empirical lake volume estimates and identifying critical depths for future hazard and risk reduction". Surge velocity is not mentioned – which is not a problem in itself, as other papers have investigated this at the same lakes (e.g., Bazai et al. 2022, Li et al. 2023). The authors themselves warn that this part of their results should be treated tentatively and with caution. In my opinion, the more valuable results are the surprisingly accurate geometric estimates of lake volumes and the investigation of critical lake depths. I would therefore agree with the other reviewers in suggesting that the part on surge velocity be omitted.

Thank you for these comments. We agree that the data relating surge velocity and GLOF volumes is sparse, which is why we present a cautious interpretation. However, as we noted in the last round of

reviews, we believe that it is important to report this relationship. This section is central to our arguments because, as we had already indicated in the text, there is a direct relationship between the negative function between surge velocity and the volume of the lake that maps on to geometric changes in lake volume including the dimension of lake depth. Going over the original reviews, only the second reviewer suggested we might omit this section and the current reviewer (in a comment below) makes suggestions as to how we can strengthen this section. We have provided additional explanation for retaining this section in reply to a comment below. Rather than delete the section we have provide an improved justification for this aspect of the study within the Introduction.

Specific comments:

1)Chapter 2, Study Area, is a welcome addition to the manuscript and serves its purpose well. However, it lacks a review of previous research for the selected glaciers and their lakes. There has been some research focused on these three glaciers and their lakes, some even focusing on the relationship between surge velocity and GLOF occurrence. For example, Round et al. (2017) state that surges are the main factor in lake formation at Kyagar Glacier, while Li et al. (2023) found that the volume of the lake there reaches its maximum three years after a surge period. Furthermore, Bazai et al. (2023) conclude that surge velocities have a significant impact on lake formation at Khurdopin. I would therefore encourage the authors to add some paragraphs detailing previous research in order to avoid duplication in later parts of the manuscript.

Citing of key references later in the text is always for a given context so there is no duplication. However, we take the point that a little more could be added to the Study Area section to provide context. To that end we have added text at line 120 as follows: "For example, Round et al. (2017) concluded that surges were the main factor controlling the formation of ice-dammed lakes associated with the Kyagar Glacier, and li et al. (2023) states that the volumes of the lakes reaching a maximum three years after the surge period. Similarly, Bazai et al. (2023) concluded that surge velocities have a significant effect of lake formation related to the Khurdopin Glacier."

Regarding Figure 1, I have some minor annotations, mainly for the overview panel in the top right: The fonts are too small to read – and not really necessary, as the other regions of HMA are not part of this study. Also, the long-lat grid is too small, and also not necessary for this kind of map. Thirdly, I understand that the map depicts a hillshade of the region, but the color map is unusual and would need to be explained. I suggest using a surface reflectance with low opacity and an underlying hillshade would be easier to understand, but that my only be my opinion.

2) In the three small panels, the authors indicate a "Dam site". Maybe I'm misunderstanding something, but isn't the location of the ice dam at the very other end of the lake where the lake connects to the glacier? The arrows seem to point to the inlet of the lake.
At last (and again, this is just personal preference): Please make the outline of the three glaciers stand out from the outlines of all other glaciers. If you choose to include the Indus, please use a shapefile with a higher resolution. And to be *very* nitpicky: Please adjust the scale bar lengths to 100 km (panel a) and 1000km (overview panel).

Thank you for your detailed feedback on Figure 1. We have carefully considered all of your suggestions and have made the following revisions in the updated figure:

**Overview Panel:** We have removed the font labels and the lat-long grid, as they were not necessary for this map and could detract from the clarity.

**Hillshade and Color Map:** We adjusted the map by using a surface reflectance layer with low opacity and an underlying hillshade, as per your suggestion. This should improve the visual clarity and ease of interpretation.

**Dam Site Label:** We reviewed the "Dam site" locations and corrected the placement to accurately indicate the position of the ice dam at the lake's connection with the glacier, as the previous arrows were mistakenly pointing toward the inlet.

**Glacier Outlines:** We have now highlighted the outlines of the three glaciers central to our study, distinguishing them from other glaciers for better emphasis.

**Indus River Shapefile:** We have replaced the previous shapefile with a higher resolution one to improve the depiction of the Indus River.

**Scale Bar:** We have adjusted the scale bar lengths to 100 km for panel (a) and 1000 km for the overview panel, as per your suggestion.

We appreciate your thoughtful suggestions, which have helped enhance the clarity and precision of the figure.

3) In Chapter 3.3 (L197f), the authors claim that the inlet of the subglacial conduit is always located at the deepest position in the ice dam. This may well be true for most cases, but the possibility of higher drainage channels in the ice should be mentioned, as this would have a large influence on the potential GLOF volume. I would also advise the authors to rephrase the start of this paragraph (L195) because, contrary to their statement, there are records of higher drainage channels at Kyagar Glacier (Li et al. 2023).

Thank you for making this point. We have added the following text at new line c. 203 "For the Kyager Glacier, Li et al. (2023) suspected that the drainage conduit may not have been at the deepest part of the lake basin and its configuration changed between GLOF events."

4) As mentioned in the general comments: Chapter 4.1 is very long, although the reviewers have criticized the presentation of the surge-GLOF relationship. Although the authors have partially addressed this, the chapter still adds little to the scientific discourse. The results presented are tentative at best, but there are other studies that clearly define the relationship between surge events and lake development as an increase in lake volume in the months/years after surges, i.e., a negative relationship between velocity and volume. The focus of this paper should be on estimating of lake volume and critical water pressure/lake depth, not on the relationship between surge velocity and lake formation. I agree with the other reviewers in suggesting that the paper be restructured and the surge aspect de-emphasized in order to properly focus on the more original content.

Thank you for your comments on Chapter 4.1. We appreciate your feedback and understand your concerns regarding the length of the section and its focus on the surge-GLOF relationship. However, we would like to clarify our reasoning for including this discussion and argue that the findings presented offer valuable contributions beyond what has been covered in previous studies.

It is true that studies, such as those by Round et al. (2017) and Li et al. (2023), have discussed the relationship between glacier surges and lake formation, specifically for the Kyagar Glacier. Both studies highlight that glacier surges are indeed responsible for the formation of lakes. For example, Round et al. (2017) suggests that historic lake outbursts could indicate surge activities at Kyagar Glacier, while Li et al. (2023) identifies surge events as a key factor in lake formation and further notes that the Kyagar Glacier lake volume reaches its maximum approximately three years after a surge.

However, what distinguishes our study is the more in-depth exploration of the relationship between glacier velocity and lake volume, along with the mechanisms leading to glacier lake outburst floods (GLOFs). Previous research has not fully explored how glacier velocity controls lake volume nor delved into the timing and triggers of GLOF events. Our study provides a more comprehensive

analysis that includes data not only from Kyagar Glacier but also from Khurdopin and Shishper Glaciers. By examining multiple glaciers, we aim to deepen the understanding of the surge-GLOF relationship, with a particular focus on estimating lake volume, critical water pressure, and lake depth—factors that are crucial for GLOF hazard anticipation.

While we acknowledge the suggestion to shift focus solely to estimating lake volume and critical water pressure/lake depth, we believe that the inclusion of the surge-velocity discussion offers a necessary foundation for understanding the dynamics of lake formation and the conditions that lead to GLOFs. Thus, we argue that this section of the paper is essential to support the original content and the broader context of GLOF hazards in the Karakoram region.

To address your concern about Chapter 4.1's length, we must say that it is not very long, totalling only 569 words! Much of the physical length in the manuscript from is taken up by Figs 4 and 5. Fig. 5 which you suggest (see other comment) could be moved to the Supplement was suggested as a necessary addition to the manuscript by another referee. We must agree with him/her that the whole text is easier to understand given the process controls that are shown now in Fig. 5.

5) The conclusion is indeed very short, as previously mentioned by the reviewers. Furthermore, the first five lines are not really part of the conclusion but more of a secondary introduction to the topic. To my mind, the conclusion should comprise the summary of the findings (a little bit more detailed than in its current state) and give indications on how to use them (which are currently absent). The ability to apparently determine the lake volume quite accurately based on the geometric estimate is an important point and should be the focus of the paper, together with the critical depth parameter. In the conclusion, the authors should describe in more detail how and when these values can be applied meaningfully.

Thank you for bringing this matter to our attention. We agree with you. We have deleted the first two sentences and added text to indicate how our findings might be used in a practical sense.

Technical corrections:
L 165: "Khurdophin" → "Khurdopin"

Corrected

L 206: The layout of Table 1 seems very crowded. If the need arises, the column "Type of drainage" could be omitted, because it mirrors the values in "Vol after".

Agreed. Column has been deleted

L 261: The assumptions are not addressed in the Discussion, but in the Results.

Corrected to 'Results'

L 264: "ice dam volume" → "ice-dammed lake volume"

Corrected

L 281: For the sake of completeness some comments on Figure 4, despite my concerns with the whole surge analysis in general and chapter 4.1 more specifically: My main point is that panel d does not correctly summarize the other three panels. The bars representing the surge periods do not correspond to the position of the GLOF events. One example: The first four GLOF events at Khurdopin appear during the surge period in panel a, but in panel d they all appear after it. The same applies for multiple other occasions. Also, the labeling of the X-Axes is still confusing, as the labels area sometimes five and sometimes six years apart – in the same panel. Why is the X-Axis in panel d formatted differently

when the style of the other panels could be used here as well? Why doesn't the X-Axis in Panel c start at 0 as well?

Thank you for your insightful comments regarding Figure 4. We have carefully reviewed the issues you raised, particularly concerning panel d.

**Panel d:** We have rearranged the data in panel d for all three glaciers. Initially, we presented the surge period only until the peak velocity. Thanks to your observation, we have now extended panel d to cover the entire surge period, which better aligns with the GLOF events and ensures consistency with the other panels.

**X-Axes:** The X-Axis labels were set at varying intervals due to data limitations, which is why they appear five to six years apart in some sections. After careful consideration, we have maintained this approach to preserve data integrity.

**Panel c:** We have updated the X-Axis of panel c to start at 0, as per your suggestion, to improve clarity.

We appreciate reviewer attention to these details, which has significantly improved the figure's accuracy and presentation.

L 301: This figure is a response to a reviewer's comment, but since it takes up a lot of space, I would suggest moving it to the supplementary material.

We appreciate that the figure takes-up space. However, we strongly disagree with moving it to the Supplement. The dynamics of damming and GLOF draining are complicated. This figure which, as the reviewer notes, was suggested as an addition by a previous reviewer, we think will help readers understand the time sequence of the processes we are reporting. Consequently, we have retained it.

L 318: In this chapter and the next, the authors assume the lake surface to be a triangle. This is only the case for the lake at Shishper Glacier. I would encourage the authors to address the question how their method can be applied to more unregular surfaces, like the lakes at Khurdopin and Kyagar.

This comment is perplexing. We assume the lake surface area is a triangle for all the three lakes. The degree of fit is reasonable in the case of Khurdopin as well as Kyager and Shishper, as is shown in Fig. 7. Clearly if a lake might conform to some other geometric shape, then the triangular approximation would not be appropriate. We provide an alternative volume calculation based on an irregular pentahedron as well as two references that consider other lake shapes.

L 335: In the corresponding figure, this coefficient is given as 0.93. Which is correct?

The reviewer is referring to the wrong panel (i.e. panel b). The correct coefficient in both panel a and in the main text is 0.88

L 347: "of the lakes was determined…" → "of the lakes determined…"

We have not made this correction as the text as written is grammatically correct and the suggested change is not grammatically correct.

L 349: The authors need to provide references for their statement that the lake depth is the main driver for tunnel drainage. According to other studies, low water pressure inside or under the glacier can also trigger a drainage.

We have added a reference here as the referee suggests. As was made clear from the sentence that follows this statement, the connectivity has to be made within the subglacial conduit. We have edited the text at this point and added three references to make it clear why the lake depth is regarded as a main driver.

L 355: The authors should choose their words more carefully when describing the applicability of their n'-value. It could play a valuable role in determining GLOF probability, but while they do not say so explicitly, readers could gain the impression that with a lake level of n'=0.6 there is the danger of a GLOF occurring imminently. It should be made clearer that a GLOF warning would be based on more components than lake depth.

In the original text we state "Thus, $n' = 0.60$ can be regarded as a warning level value with the potential for a GLOF occurring imminently increasing as $n'$ approaches unity". Thus, we do believe that there is an imminent risk of a GLOF. We do not agree that a warning would only be made after other components are considered. We thought about inserting additional text at this point to indicate that consideration of other parameters would be wise, but that takes the text off into a direction whereby the alternative indicators would have to be spelt out and discussed. We rather keep the text as it is, so that authorities can considering issuing a warning when $n' = 0.60$.

L 363: Please be more specific here: Low values of n' do not necessarily indicate shallow lakes, as it is a relative value. A large lake with a maximum depth of 300m could still contain a significant amount of water with a value of n'=0.5.

We do not state that low values of n' necessarily indicate shallow lakes. We have edited the text to indicate that this statement only refers specially to the lakes we have studied.

L 363: As one reviewer mentioned before: The reasoning behind the 500kPa is not clear and may confuse the reader. You have three glaciers with three significantly different water pressures – why choose the Kyagar? Do you mean to suggest that lakes shallower than 50m (~500kPa) are not capable of producing significant GLOFs?

The reason for selecting 500kPA is that it is the minimum we recorded for the three glaciers, and this is stated in the text. Just above this sentence we have indicated that shallower lakes are of minimal risk. By modifying the text, we have also indicated that this pressure value applies to GLOFs in the region, which indicates to the reader that the value may not apply to other regions of the world.

L 364: I agree with the other reviewers: The regional aspect of the values for depth and water pressure is not clear. Why is the region mentioned here?

'region' is mentioned here as a caveat, so that readers do not necessarily think the value applies across the world. We have added a further qualifier here in the form of an additional sentence: "However, consideration should be given to local conditions when applying the findings of this study to other locations around the globe."

L 391: Again – why are your results specific to the Karakoram region? Even the three glaciers selected for this study are very different in terms of the shape, area, depth, and volume of their lakes – which of these characteristics are typical for this region?

From the revised text given above to answer the previous two queries it is evident that we are adding a caveat that the findings of this regional study do not necessarily apply around the globe. We do not argue that specific aspects of these lakes such as shape, area etc are typical of the region. We have just studied three glaciers within the Karakorum that are associated with GLOFs and extracted some commonality in behaviour.

L 419: The value of 0.6 appears solid in this context, but the second part of this sentence should be rephrased: the lake depth would be part of the hazard determination, and in itself not sufficient to issue a warning.

The text at this point states "From this result, we infer that a safe lake level can be defined as < 0.60, while the trigger level is ≥ 0.60.". The reviewer states that the first clause is acceptable but not the second. However, the second clause follows automatically once the first clause is accepted as true. The reviewer does not indicate why using the critical depth alone as a GLOF warning is not sufficient. For reasons of safety, one might use the critical depth alone, rather than spend time considering other indicators as we have noted in response to a query above.

L 422-424: This should be mentioned before and not be part of the discussion, as it does not discuss results.

We added this text at this point to pull the discussion together to a concluding point. We have deleted it at this location and moved it to the Conclusions where it has more impact.

L 451: A small point: n'=0.6 does not automatically mean 510kPa, as n' depends on the water depth, while 510kPa corresponds to a fixed depth of approximately 50m. If n' is only valid from a water depth of 50m, this should be mentioned in the paper.

This is a good point. We have added the following caveat to the Conclusions. "In this respect, it should be noted that the water pressure recommended herein as potentially of concern (510kPA) pertains to lake depths of c. 50 m, whereas deeper lakes in other regions might drain at different values of pressure."

References:

Bazai, N. A. et al. (2022) Glacier surging controls glacier lake formation and outburst floods: The exam-ple of the Khurdopin Glacier, Karakoram, Glob. Planet. Change, 208. 10.1016/j.glopla-cha.2021.103710.
Li, G. et al. (2023) Characterizing the surge behaviour and associated ice-dammed lake evolution of the Kyagar Glacier in the Karakoram, The Cryosphere 17(7), 10.5194/tc-17-2891-2023.
Round, V. et al. (2017) Surge dynamics and lake outbursts of Kyagar Glacier, Karakoram, The Cry-osphere 11(2), 10.5194/tc-11-723-2017.